

**Climatic and extreme weather variations over Mountainous Jammu and Kashmir,**
**India: Physical explanations based on observations and modelling**
Sumira N. Zaz[1], Romshoo Shakil Ahmad[1], Ramkumar Thokuluwa Krishnamoorthy[2*], and
Yesu Babu Viswanadhapalli[2]
1.   Department of Earth Sciences, University of Kashmir, Hazratbal, Srinagar,
Jammu and Kashmir-190006, India
2.   National Atmospheric Research Laboratory, Dept. of Space, Govt. of India, Gadanki, Andhra
Pradesh 517112, India
**Email:** zaz.sumira@gmail.com, shakilrom@kashmiruniversity.ac.in, tkram@narl.gov.in,
yesubabu@narl.gov.in;
***Corresponding author** (tkram@narl.gov.in)
**Abstract**
The Himalaya is very sensitive to climatic variations because of its fragile environmental and
climatic settings. There are clear and strong indicators of climate change reported for the Himalaya,
particularly the Jammu and Kashmir region in the western Himalayas. In this study, the detailed
characteristics of long and short term as well as localized variations of temperature and precipitation are
analysed for six meteorological stations (Gulmarg, Pahalgam, Kokarnag, Quazigund, Kupwara and
Srinagar) over Jammu and Kashmir, India for a period of 37 years during 1980-2016 by making use of
observed stations data, WRF model downscaled monthly-averaged surface temperature and precipitation
and ERA-interim (ERA-I) reanalysis data. . The annual and seasonal temperature and precipitation changes
were analysed by carrying out the Student's t-test, Mann-Kendall, Spearman Rho and Cumulative deviation
statistical tests. The results show an increase of 0.8°C in average annual temperature over thirty years
during 1980-2016 with higher increase in maximum temperature (0.97°C) compared to minimum
temperature (0.76°C). Analyses of annual mean temperature at all the stations reveal higher rise at high-
altitude stations of Pahalgam (1.13°C) and Gulmarg (1.04°C) at the confidence level of S=99%.
Precipitation patterns in the valley show slight decrease in the annual precipitation at Gulmarg and
Pahalgam stations at the confidence level of S=90%.Seasonal analyses show increase in the winter and
spring temperature at all stations at the confidence level of S=95%with prominent decrease in spring
precipitation at S=99%. The present study reveals that variation in temperature and precipitation during
northern winter (December - March) has close association with the North Atlantic Oscillation (NAO).
Further, the observed temperature data (monthly averaged data for 1980-2016) at all the stations shows
good correlation of 0.86 with the results of WRF and therefore the model downscaled simulations can be



considered as a valid scientific tool for climatic change studies in this region. Using ERA-I potential vorticities in the upper troposphere over the Jammu and Kashmir region, it is found that the extreme weather event of September 2014occurred due to the breaking of intense Rossby wave activity over Kashmir. As the wave could drag lots of water vapour from both the Bay of Bengal and Arabian Sea and dump them in the region through wave breaking, resulting in the historical devastating flooding of the whole Kashmir valley in the first week of September 2014 accompanied by the extreme rainfall events measuring more than 620 mm in some parts of the Pir Panjal range in the South Kashmir.

## 1.   Introduction

Climate change is a real Earth's atmospheric and surface phenomenon and the influences of which on all the spheres of life are considered significant everywhere in the world at least in the past few decades. Extreme weather events like anomalously large floods and unusual drought conditions associated with climate change play havoc with livelihoods of even established civilizations particularly in coastal and high-mountainous areas. Jammu and Kashmir, India, located in the Western Himalayan region, is one such cataclysmic mountainous region where the significant influence of climate change on local weather has been observed for the last few decades;(1) shrinking and reducing glaciers,(2) devastating floods, (3) decreasing winter duration and rainfall, (4) increasing summer duration and temperature etc.(Solomon et al., 2007; Kohler and Maselli 2009; Immerzeel et al., 2010;Romshoo et al., 2015; Romshoo et al., 2017). Western disturbances (WD) is considered as one of the main sources of precipitation(in the form of rainfall/snowfall)for the Jammu and Kashmir region, which brings water vapour mainly from the tropical Atlantic Ocean, Mediterranean Sea, Caspian Sea and Black sea. The Indian south-west and north-east monsoons are other important sources during Northern summer and winter seasons respectively. Though WD is perennial, but it is most intense during northern winter (December-February). Planetary-scale atmospheric Rossby-waves have potential to significantly alter the distribution and movement of WD according to their intensity and duration (few to tens of days). Since WD is controlled by planetary-scale Rossby waves in the whole troposphere of the subtropical latitude region, diagnosing different kinds of precipitation characteristics is easier with the help of potential vorticity (PV) at 350K potential temperature (PT) and 200 mb level pressure surface(PS) as they are considered as proxies for Rossby wave activities (Ertel, 1942; Bartels et al., 1998).For example, Postel and Hitchman (1999) studied the characteristics of Rossby wave breaking (RWB) events occurring at 350K PT surface transecting the subtropical westerly jets. Similarly, Waugh and Polvani (2000) studied RWB characteristics at 350K PT surface in the Pacific region during northern fall–spring with emphasis on their influence on westerly ducts and their intrusion into the tropics. Since PV is a conserved quantity on isentropic and isobaric surfaces (ISOES & ISOBS), it is widely used for investigating large-scale dynamical processes associated with frictionless and adiabatic



flows. Moreover, all other dynamical parameters, under a given suitable balanced-atmospheric-background
condition, can be derived from PV and boundary conditions [Hoskins et al., 1985].

Divergence of the atmospheric air flows near the upper troposphere is larger during precipitation,

leading to increase in the strength of PV. Because of which generally there will be a good positive
correlation between variations in the strength of PV in the upper troposphere and precipitation over the
ground provided that the precipitation is mainly due to the passage of large-scale atmospheric weather
systems like western disturbances, monsoons etc. Wind flows over topography can significantly affect the
height distribution of water vapour and precipitation characteristics.. Because of this, one can expect that
the positive correlation between variations in PV and precipitation be modified significantly depending
upon both the topography and wind flow strength. These facts need to be taken into account while finding
long-term climatic variations of precipitation near mountainous regions like the western Himalaya. The
interplay between the flow of western disturbances and topography of the western Himalaya can  further
complicate the identification of source mechanisms of extreme weather events like the ones that occurred in
the western Himalayan region;2014 Kashmir floods, 2010 Leh floods, in the Jammu and Kashmir region
and 2013 in the Uttrakhand  region. This necessitates making use of the proper surrogate parameters like
PV and distinguishing between different source mechanisms of extreme weather events associated with
both the long-term climatic impacts of remote origin and short-term localized ones like organized
convection.

The main aim of the present study is to investigate the climatic variation of surface temperature

and precipitation over the Jammu and Kashmir, India region of the western Himalayas in terms of
atmospheric Rossby wave activity in the upper troposphere. Since PV is considered as a measure of Rossby
wave activity, the present work analyses in detail, for a period of 37 years during 1980-2016, monthly
variation of PV (ERA-interim reanalysis data, Dee et al., 2001) in the upper troposphere (at 350 K potential
temperature and 200 mb pressure surfaces) and compares it with observed  surface temperature and rainfall
(India Meteorological Department, IMD) at six widely separated mountainous locations with variable
orographic features (Srinagar, Gulmarg, Pahalgam, Qazigund, Kokarnag and Kupwara). There exist several
reports on climatological variation of hydro-meteorological parameters in various parts of the Himalayas.
For example, Kumar and Jain (2009) and Bhutiyani et al. (2010) found an increase in the temperature in the
north-western Himalayas with significant variations in precipitation patterns. Archer and Fowler (2004)
examined temperature data of seven stations in the Karakoram and Hindu Kush Mountains of the Upper
Indus River Basin (UIRB) in search of seasonal and annual trends using statistical test like regression
analysis. Their results revealed that mean winter maximum-temperature has increased significantly while
mean summer minimum-temperature declined consistently. On the contrary, Lui et al (2009) examined
long-term trends in minimum and maximum temperatures over the Tibetan mountain range during1961-
2003 and found that minimum temperature increases faster than maximum temperature in all the months.





Romshoo et al. (2015) observed changes in snow precipitation and snow-melt-runoff in the Kashmir valley
and attributed the observed depletion of stream flow to the changing climate in the region. Bolch et al.
(2012) reported that the glacier extent in the Korakaram range is increasing.

These contrasting findings of long term variations in hydro-meteorological parameters in the
Himalayas need to be verified by analyzing more historic climatic data available in the region. However,
the sparse and scanty availability of regional climate data poses challenges in understanding the complex
microclimate in this region. Therefore, studying the relationship of  recorded regional (Jammu and
Kashmir) climatic variations in weather parameters with remote and large-scale weather phenomena such
as the North Atlantic Oscillation (NAO), and El Niño Southern Oscillation (ENSO)become-necessity for
understanding the physical processes that control the locally observed variations (Ghashmi, 2015).Archer
and Fowler(2004) and Iqbal and Kashif (2013) found that large-scale atmospheric circulation like NAO
influences significantly the climatic condition of Himalayas. However, detailed information about variation
in temperature and precipitation and its teleconnection with observed variations of NAO is inadequately
available for this part of the Himalayan region (Kashmir Valley).

## 127 2. Geographical setting of Kashmir


The inter mountainous valley of Kashmir has unique geographical setting and  it is located
between the Greater Himalayas in the north and Pir Panjal ranges in the south, roughly within the latitude
and longitude ranges of $33^0$ 55′ to $34^o$ 50′ and 74°30′ to $75^o$ 35′ respectively (Fig.1). The heights of these
mountains range from about 3,000 to 5,000 m and the mountains strongly influence the weather and
climate of the region. Generally the topographic setting of the six stations, though variable, could be
broadly categorized into two; (1) stations located on plains (Srinagar, Kokarnag, Qazigund and even
Kupwara) and (2) those located in the mountain setting (Gulmarg, Pahalgam). Physiographically, the valley
of Kashmir is divided into three regions; Jhelum valley floor, Greater Himalayas and Pir Panjal. In order to
represent all the regions of the valley, six meteorological stations located widely  with different mean sea
levels (msl), namely, Gulmarg (2740m), Pahalgam (2600m) Kokarnag (2000m), Srinagar (1600m),
Kupwara (1670m) and Qazigund (1650m)were selected for analyses of observed weather parameters. The
topographical nature of the surroundings of these six stations (Fig. 1) is given below.
1. Kupwara:  Located on plane surface bounded on three sides by mountains.
2. Pahalgam: Located on mountain top
3. Kokarnag: Located onplane surface.
4. Srinagar:   Located on plane surface in an urbanized area
5. Gulmarg:  Located on mountain top
6. Qazigund: Located on plane surface.



The Kashmir valley is one of the important watersheds of the upper Indus basin harbouring more
than 105glaciers and it experiences the Mediterranean type of climate with marked seasonality (Romshoo
and Rashid, 2014). Broadly, four seasons (Rashid et al., 2015) are defined for the Kashmir valley; winter
(December to February), spring (March to May), summer (June to August), and autumn (September to
November).The annual temperature in the valley varies from about -10°C to 35°C. The rainfall pattern in
the valley is dominated by winter time precipitation associated with western disturbances (Dar et al., 2014)
while the snow precipitation is received mainly in winter and early spring season (Kaul and Qadri, 1979).

## 3. Data and Methodology


India Meteorological Department (IMD) provided 37 years (1980-2016) of data of daily
precipitation, maximum temperature and minimum temperature for all these six stations. Monthly averaged
data were further analysed to find long term variations of the weather parameters. Statistical tests including
Mann-Kendall, Spearman Rho, Cumulative deviation, Student's t-test were performed to determine long
term trends and turning point of weather parameters with statistical significances. Similar analyses and tests
were performed also for the Weather and Research Forecasting (WRF) model simulated and ERA-Interim
reanalyses data (0.75° by 0.75° spatial resolution in the horizontal plane) of same weather parameters and
for the NAO index. Brief information about these datasets is provided below.

### 3.1. Observational and model datasets used in this study

The obtained observational data were carefully analysed for homogeneity and missing values.
Analyses of ratios of temperature from the neighbouring stations with the Srinagar station were conducted
using relative homogeneity test (WMO, 1970). It is found that there is no significant inhomogeneity and
data gap for any station.  Few missing data points were linearly interpolated and enough care was taken not
to make any meaningful interpretation during such short periods of data gap in the observations. Annual
and seasonal means of temperature and precipitation were calculated for all the stations and years. To
compute seasonal means, the data were divided into the following seasons: winter (December to February),
spring (March to May), summer (June to August) and autumn (September to November).Trends in the
annual and seasonal means of temperature and precipitation were determined using Mann–Kendall (non-
parametric test) and linear regression tests (parametric test) at the confidence levels of S=99%, S=95% and
S=90%. These tests have been extensively used in hydrometeorological data analyses as they are less
sensitive to heterogeneity of data distribution and least affected by extreme values or outliers in data series.
Various methods have been applied to determine change points of a time series (Radziejewski et al., 2000;





Chen and Gupta, 2012). In this study, change point in time series of temperature and precipitation was
identified using cumulative deviation test (Pettitt, 1979). This method detects the time of significant change
in the mean of a time series when the exact time of the change is unknown (Gao et al., 2011).

The data of winter NAO index during 1980–2010from Climatic Research Unit were obtained for

analyses from the web link htpp://cru@uea.ac.uk. The winter (December - March) NAO index is based on
the difference of normalized sea level pressure (SLP) between Lisbon, Portugal and Iceland, which is
available from 1964 onwards. Positive NAO index is associated with stronger-than-average westerlies over
the middle latitudes (Hurrell, 1997).  Correlation between climatic variations of mean (December-March)
temperature, precipitation and NAO index was determined using Pearson correlation coefficient method.
To test whether the observed trends in winter temperature and precipitation are enforced by NAO, linear
regression analysis was performed.

**3.2. WRF Model configuration**

The Advanced Research WRF version 3.9.1 model simulation was used in this study to downscale

the ERA-Interim (European Centre for Medium Range Weather Forecasting ReAnalysis) data over the
Indian Monsoon region. The model is configured with 2two-way nested domains (18 km and 9-km
horizontal resolutions), 51 vertical levels and model top at 10 hPa level. The first domain of the model
extends over the whole Indian monsoon region (28°E to 112°E and 20°S to 45°N) with 18-km horizontal
grid resolution while the second domain (9-km resolution) covers the Indian sub-continent region.The
initial and boundary conditions supplied to WRF model are obtained from ERA-Interim 6-hourly data.
Model physics used in the study for boundary layer processes is Yonsei University's non-local diffusion
scheme (Hong et al., 2006), the Kain-Fritsch scheme for cumulus convection (Kain and Fritsch, 1993),
Thomson scheme for microphysical processes, the Noah land surface scheme (Chen and Dudhia, 2001) for
surface processes, Rapid Radiation Transfer Model (RRTM) for long-wave radiation (Mlawer et al., 1997),
and the Dudhia (1989) scheme for short-wave radiation. The physics options configured in this study are
adopted based on the previous studies of heavy rainfall and Monsoon studies over Indian region (Srinivas
et al., 2013, Madala et al., 2016, Priyanka Ghosh et al, 2016; Srinivas et al., 2018).


For the present study, the WRF model is initialized on daily basis at 12 UTC using ECMWF ERA

interim data and integrated for a complete 36-hour period using the continuous re-initialization method (Lo
et al., 2008 and Viswanadhapalli et al., 2017). Keeping the first 12-hours as model spin-up time, the
remaining 24-hour daily simulations of the model are merged to get the climate data during 1980-2016.To
find out the skill of the model, the downscaled simulations of WRF model are validated at six IMD surface





meteorological stations. The statistical skill scores such as bias, mean error (ME) and root mean square
error (RMS) were computed for both the simulated temperature and observed temperature data of IMD.

## 4. Results and Discussion:

### 4.1. Trend in annual and seasonal temperature


From Table 1, it is evident that there is an increasing trend at different confidence levels in annual and
seasonal temperatures of all the six stations (Pahalgam, Gulmarg, Kokarnag, Srinagar, Kupwara and
Qazigund), located indifferent topographical settings. Higher value of statistical significance between
Mann-Kendall and linear regression test results is considered here. In Table 4, these test results are
provided in detail along with Student's t test results for comparison. It is to be noted here that the latter test
is not recommended in general for trend analyses as it could not identify properly odd data points. During
1980-2010, Pahalgam and Gulmarg, located at higher elevations of about2500m amsl (above mean sea
level), registered an increase in average annual temperature by 1.13°C and 1.04°C respectively at the
confidence level of S=99%(Fig. 2a). "S in S=99%" indicates statistically significant. It is to be noted that
hereafter it will not be mentioned explicitly about the period 1980-2010 and the statistical significance of
derived values. All the results are subjected to statistical tests with confidence level of statistical
significance at S=99% unless otherwise mentioned explicitly. Further, values denoting "less than" refer to
S=99% and the confidence levels corresponding to given values are provided within closed brackets.
Kokarnag and Kupwara, located at the heights of about 1800-2000m amsl, showed an increase of 0.9°C and
1°C respectively at S=99% (Fig. 2a). However, Srinagar and Qazigund, located at the heights of about
1700m-1600m amsl, exhibited an increase of 0.65°C and 0.44°Crespectively at S=99%(Fig. 2a).

The analysis of maximum and minimum temperatures (Table 1 and Fig. 2b) for these six stations
reveals higher increase in maximum temperature. At S=99%, Pahalgam and Kupwara recorded the highest
rise of ~1.3°C followed by Kokarnag (1.2°C), Srinagar (1.1°C) and Qazigund (0.6°C). The exception is that
Gulmarg (being a hilly station) shows less than 0.6°C in maximum temperature (0.6°C at S=90%) and
higher increase of 1.21°C in minimum temperature.. The minimum temperature shows lowest increase of
0.3°Cat Srinagar (Fig. 2c). Analyses of seasonal mean minimum and maximum temperatures in the valley
reveal higher increase in maximum temperature in winter and spring seasons. Among four stations
(Gulmarg, Pahalgam, Kokarnag and Kupwara), the mean winter temperature of Gulmarg indicates an
increase of less than 1°C (1°C at S=95%)while Pahalgam, Kokarnag and Kupwara shows an
increase of 0.9°C and less than 0.9°C (0.9°C at S=90%) respectively(Table 1and Fig. 2d).On the
contrary, Qazigund and Srinagar showed a slight increase of less than 0.4°C and 0.5°C (S=90%)
respectively. Mean spring temperature shows higher rise comparing to other seasons for all the



stations. While Gulmarg shows an increase of less than 1.4°C ( S=95%) ,Pahalgam, Kupwara and
Kokarnag reveals an increase of 1.3°C.and Srinagar and Qazigund display an increase of less than 1°C and
0.6°C (S=95%)respectively as shown in Table 1and Fig. 2e. In summer, the temperature rise for Pahalgam
is about less than 0.6°C (S=90%) and for Kokarnag and Gulmarg, it is about  0.4°C (insignificant level
(NS), Table 1).Kupwara, Qazigund and Srinagar  reveal an increase of less than 0.3°C, 0.2°C and 0.1°C
(S=90%) respectively (Fig, 2f). In Autumn, Gulmarg shows an increase of 0.9°C while Pahalgam and
Kupwara shows less than 0.6°C (S=95%). On the contrary, Kokarnag and Qazigund shows less than 0.4°C
(S=90%) but Srinagar shows no significant increase (Fig. 2g and Table 1).

**4.2. Trend in annual and seasonal precipitation**

The annual precipitation pattern of the valley is comparable to that of temperature with higher

decrease observed at the upper elevation stations of Gulmarg and Pahalgam. These two stations show an
average decrease in annual precipitation at S=95% and S=90% respectively (Fig. 3a and Table 2). Similar
to temperature, Table 5 provides in detail the test results of Mann-Kendall, linear regression and Student's
t. Kokarnag and Kupwara show decrease at S=90%. The lower elevated stations, Qazigund and Srinagar,
exhibit decrease at S=95% (Fig.3a).The analysis of winter precipitation reveals maximum decrease at
Gulmarg and Kokarnag followed by Kupwara and Pahalgam at S=90%. On the other hand, Srinagar and
Qazigund display an average insignificant (NS) decrease (Table 2 and Fig. 3b) while the spring season
precipitation exhibits decreasing trend at S=95% at Qazigund and Pahalgam(Fig. 3c). Srinagar, Gulmarg
and Kokarnag show decrease at S=99%, S=99% and S=95% respectively. The lowest decreasing trend of
42mm precipitation during 1980-2010was observed at Kupwara at S=99% (Table 2).

During the summer months, precipitation follows the same decreasing trend but not at significant

level (NS) for Gulmarg, Kupwara, Kokarnag, Pahalgam and Srinagar (Fig. 3d, and Table 2). In addition,
Qazigund shows no trendin summer precipitation. The autumn precipitation at Pahalgam,Kupwara,
Kokarnag, Srinagar, Gulmarg and Qazigund shows an average decrease at insignificant level (NS) (Fig. 3e
and Table 2). Cumulative test was used to determine the change point of trend in the annual and seasonal
variations of temperature and precipitation. The results reveal that the year 1995 is identified as the year of
abrupt increase (change point) in temperature of the valley (Fig. 4a) and the same is identified as the year
of abrupt decrease for precipitation (Fig. 4b).





### 4.3. North Atlantic Oscillation (NAO) index and winter climatic fluctuations


288  Along with the trend analyses of temperature and precipitation, the present study also investigated
the teleconnection between the NAO and temperature and precipitation over the Kashmir valley during
northern winter (December - March). The results suggest negative and positive correlations of -0.54 and
0.68between the NAO and the winter temperature, and precipitation respectively (Fig. 4c). This would
indicate that positive phase of NAO is associated with more precipitation. This correlation study suggests
that winter precipitation and temperature has some association with the winter NAO index. Interestingly,
similar to the observed temperature and precipitation changes in Kashmir in mid-nineties, abrupt variation
in the NAO index is also identified in1995.  To test whether the trends in temperatures and precipitation are
forced by theNAO, regression analysis was performed for temperatures and precipitation data during
winter(December- March) which is depicted in the Figs. 4e and f. From the results it is clear that the
observed trends in winter and spring temperature and precipitation would have been influenced by NAO.

300  The observed annual and seasonal variation of temperature at all the six stations (Gulmarg,
Pahalgam, Kokarnag, Kupwara, Qazigund and Srinagar) is correlated with WRF downscaled simulations.
The simulations show correlation of 0.66, 0.67, 0.72, 0.62, 0.79 and 0.47for Srinagar, Gulmarg, Kokarnag,
Kupwara, Pahalgam and Qazigund respectively. The annual mean simulated temperature shows very good
correlation (0.85) with the observations (Fig.5).Gulmarg, Kokarnag and Pahalgam show higher correlation
with the simulation comparing to Qazigund, Kupwara and Srinagar. However, the root mean square error
(RMSE) analysis  shows that model simulations slightly underestimate the observed values with an average
value of -0.43°C.

### 4.4. Discussion


311  The Himalayan mountain system is quite sensitive to global climatic variations as the hydrology
of the region is mainly dominated by snow and glaciers, making it one of the ideal sites for early detection
of global warming (Solomon et al., 2007; Kohler and Maselli 2009). Various reports claim that in the
Himalayas significant warming occurred in the last century.(Fowler and Archer, 2006; Bhutiyani et al.,
2007). Shrestha et al. (1999) analysed surface temperature at 49 stations located across the Nepal
Himalayas and the results indicate warming trends in the range of 0.06 to 0.12°C per year. The
observations of the present study are in agreement with the studies carried out by Shrestha et al.(1999),
Archer and Fowler (2004) and Butiyani (2007). In the present study, it is observed that the rise in
temperature is larger  at higher altitude stations of Pahalgam (1.13°C) and Gulmarg (1.04°C) whereas
Kokarnag, Kupwara, Srinagar and Qazigund recorded a rise of 0.9°C, 0.99°C, 0.04°C, and 0.10°C





respectively with an average rise of 0.8°C during1980-2010. Liu et al. (2009) and Liu and Chen (2000)
also reported higher warming trends at higher altitudes in the Himalayan regions. Wiltshire (2013)
warned,using a climate change model, that the impacts of climate change in the future will be intense at
higher elevations and in regions with complex topography.
The noteworthy observation in the present study is that drastic and significant increase in the
temperature (change point) started in 1995. The El Nino of 1998 has been recorded in the history of the
earth as one of the strongest El-ninos that brought worldwide increase in temperature (Epstein et al., 1998).
In contrast, during the 1992Elnino period, the decrease in temperature throughout the northern hemisphere
is ascribed to the natural phenomena of volcanic eruption occurred on the Mt Pinatabu (Swanson et al.,
2009; IPCC, 2013). This event interrupted the direct sunlight to reach on the surface of the earth for about
two months.
Studies of trends in seasonal mean temperature in many regions across the Himalayas indicate
higher warming trends in winter and spring months (Shrestha et al., 1999; Archer and Fowler, 2004;
Butiyani, 2009).The seasonal difference found in the present study is consistent with other studies carried
out for the Himalayas (Archer and Fowler, 2004; Sheikh et al., 2009 and Roe et al., 2003), Lancang Valley,
China (Yunling and Yiping, 2005), Tibet (Liu and Chen, 2000) and the Swiss Alps (Beniston et al., 2010),
where almost all stations recorded higher increase in the winter and spring temperatures comparing to
autumn and summer temperatures. Recent studies found that reducing snows and shrinking glaciers may
also be one of the contributing factors for the observed higher warming, because reduction in snow and
glacier can change the surface albedo of the region, which in turn can increase the surface air temperature
(Kulkarni et al., 2002; Groisman et al., 1994). Romshoo et al. (2015) and Murtaza and Romshoo
(2016)have reported that reduction of snow and glacier cover in the Kashmir regions of the Himalayas
during the recent decades could be one of the reasons of occurrence of higher warming particularly on the
higher elevated stations of Gulmarg and Pahalgam.
In the Himalayan mountain system, contrasting trends have been noted in precipitation over the
recent decades. IPCC (2001), Borgaonkar et al. (2001), Shreshtha et al. (2000),and Archer and Fowler
(2004) observed increasing precipitation patterns over the Himalayas while Mooley and Parthasarathy
(1983) and Kumar and Jain (2009) reported large-scale decadal variation with increasing and decreasing
precipitation periods. The results of the present study indicate that decrease in annual precipitation is
slightly insignificant at all the six stations except the spring season. Increasing trend in temperature can
trigger large-scale energy exchanges that become more intricate as complex topography alters the
precipitation type and intensity in many ways. Climate model simulations (Zarenistana et al. 2014; Rashid
et al. 2015) and empirical evidence (Vose et al. 2005; Romshoo et al., 2015) also confirm that increasing
temperature results in increased water vapour leading to more intense precipitation events even when the





total annual precipitation reduces slightly. Increase in temperature therefore increases the risks of both
floods and droughts. For example, the disaster flood event of September 2014 occurred in the Kashmir
valley due to high frequency and high intense precipitation.

The North Atlantic Oscillation (NAO) is the strongest weather phenomena that occur in the
Northern hemisphere due to the difference of atmospheric pressure at sea level between the Iceland low and
the Azores high. It controls the strength and direction of westerly winds across the northern hemisphere.
Surface temperatures have increased in the Northern Hemisphere in the past few decades (Mann et al.,
1999; Jones et al., 2001), and the rate of warming has been especially high (~ 0.15°C decade$^{-1}$) in the past
40 years (Folland et al., 2001; Hansen et al., 2001). NAO causes substantial fluctuations in the climate of
the Himalayas (Hurrell, 1997; Syed et al., 2006; Archer and Fowler, 2004).Several workers found a strong
connection between the NAO and temperature and precipitation in the north-western Himalayas (Archer
and Fowler, 2004; Bhutiyani et al., 2007; Sharif et al., 2012; Iqbal and Kashif, 2013). A substantial fraction
of the most recent warming is linked to the behaviour of the NAO (Hurrell, 1997; Thompson et al., 2003).
The climate of the Kashmir Himalayas is influenced by the western disturbances particularly in winter and
spring seasons. Figs.4c&d show correlation between the winter NAO and winter temperature and
precipitation over the Kashmir region. Negative correlation (-0.54) exists between winter temperature and
winter NAO index and positive correlation (0.68) for the precipitation. Linear regression analysis was used
to determine whether the variation in temperature and precipitation during the winter months (December-
March)is forced by NAO. It is found that considerable variation in winter precipitation/temperature may be
forced by winter NAO. The weakening effect of NAO particularly after 1995 has decreased the winter
precipitation and increased winter temperature in the valley. Similarly, Bhutiyani et al. (2009) and Dimri
and Dash (2012) also detected a statistically substantial decreasing trend in the precipitation pattern and
identified considerable decrease in winter precipitation which they related to weakening of NAO index.
However, detailed mechanism involved in these variations requires thorough investigation.

The comparison of WRF with the observed stations data shows a significantly strong correlation
of 0.85. It is also found that the higher elevated stations show higher correlation than the lower elevated
stations of Srinagar and Kupwara; however, good correlation could result if more precise terrain
information is incorporated in the WRF model. Various researchers (e.g., Kain and Fritsch, 1990, 1993;
Kain, 2004) also found good correlation between observed and WRF simulated rainfall events. In
conjunction with large-scale features such as the NAO andENSO, it can result in large scale variability in
the climate of this region (Ogura and Yoshizaku, 1988). Further, incorporation of mesoscale
teleconnections and their associations in the WRF model can further help in understanding large-scale
weather forecasting particularly in this region.




### 4.3. Physical mechanisms of climate and weather of Jammu &Kashmir


397  Large-scale spatial and temporal variations in the meridional winds could be due to the passage of
planetary-scale Rossby waves (RW) in the atmospheric winds..When RWs break in the upper troposphere,
it could lead to vertical transport of atmospheric air between the upper troposphere and lower stratosphere
and an irreversible horizontal transport of air mass between the subtropics and extra tropics (McIntyre and
Palmer, 1983). Rossby waves have the characteristic of remaining coherent over many days and
propagating long distances of the order synoptic to planetary scales leading to tele-connection of remote
atmospheres of global extent.  It is clear from the studies by Chang and Yu (1999) that during northern
winter months of December–January–February, Rossby wave packets can be most coherent over a large
distance of from the northern Africa to the Pacific through the southern Asia. There are reports on extreme
weather events connected to Rossby waves of synoptic to planetary scales in the upper troposphere (Screen
and Simmonds, 2014).In the northern parts of India, there is increasing trend in heavy rainfall events,
particularly over the Himachal Pradesh, Uttrakhand and Jammu and Kashmir (Sinha Ray and Srivastava,
2000; Nibanupudi et al., 2015). Long-scale Rossby waves can lead to the generation of convergence and
divergence in the upper troposphere that in turn can affect surface weather parameters like precipitation
through generation of instabilities in the atmospheric air associated with convergence and divergence
(Niranjankumar et al., 2016).

414  Using observations and MERRA (Modern-Era Retrospective Analysis for Research and
Applications reanalysis data; http://gmao.gsfc.nasa.gov/research/merra/),Rienecker et al.(2011) showed a
strong correlation between 6-10 day periodic oscillations in the upper tropospheric winds associated with
Rossby waves and surface weather parameters like atmospheric pressure, winds, temperature, relative
humidity and rainfall during a severe weather event observed at the Indian extratropical station, Nainital
(29.45° N, 79.5° E), during November–December 2011. Further they noted that when the upper
troposphere shows divergence, the lower troposphere shows convergence and as a result more moisture
gets accumulated there leading to enhancement of relative humidity and hence precipitation. It was asserted
that Rossby waves in the upper troposphere can lead to surface weather related events through the action of
convergence or divergence in the atmospheric air. It is to be noted that a passing Rossby wave can cause
fluctuations in divergence and convergence in the atmosphere at periodicities (typically 6-10 days, 12-20
days) corresponding to the Rossby waves at a particular site.

427  It was reported that Rossby waves account for more than 30% of monthly mean precipitation and
more than 60% of surface temperature over many extra tropical regions and influence short term climatic
extremes (Schubert et al., 2011). Planetary waves affecting weather events severely for long duration of the
order of months have been reported by many researchers (Petoukhov et al., 2013; Screen and Simmonds,
2014 and Coumou et al., 2014). Screen and Simmonds (2014) found that in the mid latitude regions there is



a strong association between enhanced Rossby wave activity, surface temperature and extreme precipitation
events during 1979–2012. Since slowly propagating Rossby waves can influence weather at a particular site
for long periods lasting more than few weeks, one can see the imprint of climatic variations of Rossby
waves in weather events from monthly mean atmospheric parameters.

To understand the present observation of different precipitation characteristics at all the six
mentioned stations over the study area, we compared the monthly variation of PV in the upper troposphere
with precipitations at all these stations. Potential vorticity at 350K potential temperature (PT) surface is
identified as a valuable information for investigating the activity of Rossby waves as its breakage (can be
identified through reversal of gradient in PV) at this level can lead to exchange of air at the boundary
between the tropics and extra tropics (Homeyer and Bowman, 2013). Similarly PV at 200 mb pressure
surface (PS) is more appropriate for identifying Rossby wave breaking in the subtropical regions (Garfinkel
and Waugh, 2014).

Since the Srinagar city, among the six stations, is onthe plain land with comparatively less
topographical features located in the centre of the Kashmir valley, precipitation here associated with
western disturbances is under the direct influence of planetary-scale Rossby waves. Accordingly,
correlation between PV at the 350 K PT (located near the core of the subtropical jet, Homeyer and
Bowman, 2013) and 200 mb pressure surfaces and precipitation is found significant and the correlation
becomes weaker for the other stations located at higher altitudes due to significant orographic influences.
As a result, one can see that PV (ERA-Interim data, Dee et al., 2011) in the upper troposphere varies in
accordance with precipitation, which is clearly depicted in Fig. 6, during the entire years of 1984, 1987,
1988, 1990, 1993, 1994, 1995, 1996, 1999, 2006 and 2009. One can observe that sometimes PV at 350 K
PT surface and at other times at 200 mb pressure surface follows precipitation. This would be due to the
influence of Rossby waves generated due to baroclinic and barotropic instabilities respectively.
Particularly, the correlation between PV (sometimes either one or both) and precipitation is significantly
positive during the Indian summer monsoon months of June-September of all the years from 1980 to 2009
except 1983, 1985, 1989, 2000-2005 and 2009. At present it is not known why this relation became weak
during1999-2010.

For Kokarnag (Fig. 7), topography similar to Srinagar but located in the vicinity of high
mountains, the relation between PV and precipitation particularly during the Indian summer-monsoon-
period is almost similar to that of Srinagar during 1983, 1985, 1989, 1991, 1998, 1999, 2000-2005, but in
2009 it became poor. This deterioration of the link between PV and rainfall over Kokarnag particularly
during 1999-2010 is really due to the effect of climate change, which is similar to what was observed for
Srinagar. It is intriguing that why this relation became poor during the years of 1999-2010. In the northern
Kashmir region of Kupwara (Fig. 8), msl higher by ~1 km than Srinagar, the relation between PV and



precipitation is good in the years 1982-1983, 1985-1988, 1990-1994, 1995-1996, 1999, and 2006. Similar
to Srinagar and Kokarnag, Kupwara also shows a poor link during 1999-2010. Particularly during the
summer monsoon period, the relation is good in all the years except 1989, 1998, 2000-2005, and 2009. One
interesting observation is that 1983, 1985 and 1991shows better correlation for Kupwara than Srinagar and
Kokarnag. Since Kupwara is located near elevated Greater Himalayan mountain range, Rossby waves
associated with topography would have contributed to the good correlation between PV and precipitation
here, which is not the case for Srinagar and Kokarnag.  In the case of Pahalgam (Fig. 9), located near the
Greater Himalayas, generally the link is good in almost all the years 1980-2016 but with a difference that
sometimes both the PVs and on other times only either of them follow precipitation during some months.
Particularly during summer monsoon months, similar to Kupwara, these years1989, 2000-2003, 2005 and
2009 show poor correlation. From the present observations, it can be easily ascertained that stations located
near the Greater Himalayas show similar characteristics influence dby topography-associated Rossby
waves.

For the hilly station of Qazigund (Fig. 10), located in the south Kashmir region (~3 km height)near

the foothills of Pir Panjal mountain range, the relationship is better than that observed over the northern
station Kupwara. For example, in 1988, the relation is much better over Qazigund than Kupwara. However
the opposite is true in 1987. Interestingly, in 1985, both Kupwara and Qazigund show similar variation in
PV and precipitation.   This may be due to the effect of the nature of equator ward propagation of Rossby
waves from mid latitudes. In 1995, 1997 and 1998, PV and precipitation follow similar time variation for
both Kupwara and Qazigund except for three months of January-March during which precipitation over
Qazigund but not Kupwara follows PV. Interestingly, in the whole year of 1999, precipitation at both the
stations, Kupwara and Quazigund, follows exceedingly well with PV; however in 1998, only Qazigund but
not Kupwara shows good relation. In 2009, precipitation does not follow PV for both the stations.
Interestingly in all the months of 2006, PV follows well with precipitation for both Kupwara and Qazigund.
However in September, Kupwara but not Qazigund shows good relation. In 2004, only PV at constant
potential temperature surface (350K) follows well with precipitation for both the stations. For the summer
monsoon period of June-September, these years do not show good correlation, namely, 1983, 1985, 1989,
1990, 2000-2003, 2005, 2007-2009, which is almost similar to Srinagar and Kokarnag.

In the case of Gulmarg (Fig. 11), PV and precipitation follow each other well in the years of 1988,

1993, 1994 and 1995. In 1996, during the Indian summer monsoon period of June-September, only PV at
constant potential temperature surface follows precipitation. Overall, during the summer monsoon period,
the relationship between PV and precipitation is appreciable for all the years except for 1983, 1989, 1990,
1999 and 2000-2009, which is almost similar to Kupwara and Pahalgam. It may be noted that these stations
are located near comparatively elevated mountains and hence topographically induced Rossby waves could
have contributed to this good relation. From the observations of these stations, one can come to the





conclusion easily that high altitude mountains affect the precipitation characteristics through topography
generated Rossby waves. The interesting finding here is that irrespective of the different heights of
mountains, all the stations show that during 1999-2010 the correlation between upper tropospheric PV and
rainfall became poor, indicating that some unknown new atmospheric dynamical concepts would have
played significant role in disturbing the precipitation characteristics significantly over the western
Himalayan region. This issue needs to be addressed in the near future by invoking suitable theoretical
models so that predictability of extreme weather events can be improved in the mountainous Himalaya.

During 2011-2016 (Fig. 12), it may be observed that for Gulmarg the linkage between potential

vorticities and precipitation is in general good for all these years except around July 2012, July-December
2013 and 2015. It is interesting to note here that during the historical flood event of September 2014, the
potential vorticities and precipitation follow each other but in the preceding and following years of 2013
and 2015 the linkage between PV and precipitation is rather poor as noted earlier. Similarly, all the other
stations (Srinagar, Pahalgam, Kokarnag, Kupwara, and Qazigund) also show that the link between PV and
precipitation is good around September 2014. This would indicate clearly that the extreme weather event
during September 2014 over the area occurred because of the intense large-scale Rossby wave activity and
not because of any localized adverse atmospheric thermodynamical conditions like enhanced local
convection etc. In Srinagar, most of the times PV and precipitation follow each other very well as observed
during January 2011-June 2012, January-July of 2013 & 2014,  whole 2015 and 2016. In Qazigund, this
relation is good only during January-July and September-October 2014, during the entire 2015 and 2016
(similar to Srinagar). For Kupwara, PV follows precipitation well during whole of 2011, January-July
2012, January-May 2013, January-November 2014, whole of 2015 and 2016. In the case of Kokarnag,
good relation is observed during March-August 2012, January-June 2013 and 2014, around September
2014. In contrast, the relationship is very poor in the entire year of 2015 and 2016.  Pahalgam interestingly
shows good correlation between PV and precipitation during the whole years of 2011 and 2012. In 2013,
2014, 2015 and 2016, it is good only during January-June in addition to exceptionally good near September

2014.


Finally, it may be observed that the ERA-interim reanalysis data of meridional wind velocity

(12UT) at ~3 km altitude above the mean seal level show alternating positive (southerly) and negative
values, resembling the atmospheric Rossby waves in the sub tropical region during 1-6 September 2014
(Fig. 13),. The meridional winds associated with Rossby waves could be easily noted to have their
extensions in both the Arabian Sea and Bay of Bengal, indicating that water vapour from both the regions
was attracted towards the Jammu and Kashmir, India region as the converging point of Rossby waves was
located near this region. It may be easily noticed that the waves got strengthened on 4[th] and weakened on 5[th]
and ultimately dissipated on 6[th] September. This dissipation of Rossby waves led to the dumping of the
attracted water vapour over this region leading to the historical-record heavy-flooding during this period.



This is one clear example of how synoptic scale Rossby waves can reorganize water vapour over large
scale and lead to extreme rainfall event. It is well known that subtropical westerly jet is one of many
important sources of  Rossby waves in the mid to tropical latitudes. If the subtropical jet drifts climatically
northward then the surface weather events associated with them also will drift similarly which will lead to
unusual weather changes climatically.

Interestingly from the published reports, it can be found that there is a close association between
changes in Rossby wave breaking events and climatic variations and variations in the stratospheric
dynamics (Barnes and Polvani 2013; Lu et al. 2014).Climatic merdional shift, which is in response to the
enhanced polar vortex and upper-tropospheric baroclinicity arising due to global warming, of the
tropospheric jet has been successfully linked to climatic changes in Rossby wave breaking events caused by
baroclinic instabilities (Wittman et al., 2007; Kunz et al., 2009; Rivière, 2011; Wilcox et al., 2012). The
climatic increase in the tropospheric warming arising due to baroclinic forcing of Rossby waves is more
prominent in the mid-latitudes than in the tropical regions(Allen et al., 2012; Tandon et al., 2013).This mid-
latitude warming plays an important role in driving the poleward jet shift responding to climate change
(Ceppi et al., 2014).It is to be remembered that the combined action of tropospheric baroclinic forcing
(warming) and stratospheric polar vortex can gradually move the subtropical jet from about 27° to 54°
(Garfinkel and Waugh, 2014). Using Global circulation models (GCM), linear wave theory predicts that in
response to increased greenhouse gas (GHG) forcing, mid-latitude eddy-driven jets, arising due to strong
coupling between synoptic scale eddy activity and jet streams in both the hemispheres, will be climatically
shifted poleward (Fourth report of Intergovernmental Panel on Climate Change (IV-IPCC), Meehl et al.,
2007).However, mid-latitude Rossby waves and the associated wave dissipation in the subtropical region
are predicted to move climatically equatorward due to the spherical geometry of the Earth (Hoskins et al.,
1977; Edmon et al., 1980). This propagation of location of wave breaking towards the equator will have
climatic impact on the proper relation between upper troposphere PV variations associated with Rossby
waves and the associated surface weather parameters in the subtropical latitude regions. This may be one of
the reasons that during 1999-2010, the relation between PV and precipitation became poor as observed in
the present study.

Regarding surface temperature, except for its linear long term trend, there is no clear evidence of
strong link between variations in the upper tropospheric potential vorticities and surface temperature for all
the six stations mentioned. It seems that climatic variations in the upper tropospheric vorticities have
significantly less influence on surface temperature variations.






## 5. Conclusions

Studies of climatic change in the surface temperature and precipitation over the Jammu and Kashmir, India region of the western Himalayas are carried out for a period of 37 years during 1980-2016. Analyses of the observations reveal an increase in the annual temperature by 0.8°C. Higher increase in temperature is noted for stations located at higher altitudes and that is accompanied with an insignificant decrease in annual precipitation. Long-term variation of winter temperature and precipitation has good correlation with winter NAO index. Additionally, WRF model simulations show good correlation of 0.85 with the observed data. It is found that in the recent decades, precipitation associated with both the monsoon and western disturbances has been decreasing significantly. While the monsoon deficiency is associated with decreasing difference in surface temperature between the Indian landmass and nearby Indian ocean, the deficiency associated with western disturbances during winter is due to the climatic northward displacement of subtropical jet. This subtropical jet wind helps to drag moisture associated with disturbances to the Himalayan region. Regarding historical extreme weather event associated with September 2014 floods in Jammu and Kashmir, it is found that breaking of intense Rossby wave activity over Kashmir played an important role as the wave could drag lots of water vapor from both the Bay of Bengal and Arabian Sea and dump them here through its breaking during the first week of September, 2014, leading to the extreme rainfall event measuring more than 620 mm in some parts of the South Kashmir.

## Acknowledgements:

Thanks are due to the India Meteorological Department, Pune, India, ERA-Interim reanalyses and WRF model simulation teams for the data of meteorological parameters employed in the present work. Prof. Shakil Romshoo and Dr. Sumaira Zaz gratefully acknowledge the support of the Department of Science and Technology (DST), Government of India under the research project titled "Himalayan Cryosphere: Science and Society". Dr. T. K. Ramkumar and Dr. V. Yesubabu acknowledge the support of Dept. of Space, Govt. of India. The authors express gratitude to the two anonymous reviewers and Editor for their valuable comments and suggestions on the earlier version of the manuscript that has greatly improved its content and structure.



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

**Table:**

Table 1. Annual and Seasonal temperature trend in Kashmir Valley during 1980-2010
Table 2. Annual and Seasonal Precipitation trends in Kashmir valley during 1980-2010
Table 3. Mean temperature increase at each station from 1980 to 2010
Table 4: Statistics of temperature at all the six stations of Kashmir (1980-2010)
Table 5: Statistics of precipitation at all the six stations of Kashmir (1980-2010)

**Table 1: Annual and Seasonal temperature trend in Kashmir Valley**

| Stations | Temperature Trend | Annual S% | Min S% | Max S% | Winter S% | Spring S% | Summer S=% | Autumn S=% | Abrupt Change |
|----------|-------------------|-----------|--------|--------|-----------|-----------|------------|------------|---------------|
| Gulmarg  | + | **S=99** | **S=99** | S=90 | **S=95** | **S=95** | NS | **S=99** | 1995 |
| Pahalgam | + | **S=99** | **S=99** | **S=99** | **S=99** | **S=99** | S=90 | **S=95** | 1995 |
| Srinagar | + | **S=95** | **S=95** | **S=99** | S=90 | **S=95** | NS | NS | 1995 |
| Kupwara  | + | **S=99** | **S=95** | **S=99** | S=90 | **S=99** | S=90 | **S=95** | 1995 |
| kokarnag | + | **S=99** | **S=99** | **S=99** | **S=99** | **S=99** | NS | S=90 | 1995 |
| Qazigund | + | **S=95** | S=90 | **S=95** | S=90 | **S=95** | NS | S=90 | 1995 |

+ Increase; S= significance level ; NS = Insignifance;  Bold=  high significance



**Table 2: Annual and Seasonal Precipitation trends in Kashmir valley.**

| Stations | Precipitation Trend | Annual S% | Winter S% | Spring S% | Summer S=% | Autumn S=% | Abrupt Change |
|---|---|---|---|---|---|---|---|
| Gulmarg | - | **S=95** | S=90 | **S=99** | NS | NS | 1995 |
| Pahalgam | - | S=90 | S=90 | **S=99** | NS | NS | 1995 |
| Srinagar | - | **S=95** | NS | **S=99** | NS | NS | 1995 |
| Kupwara | - | S=90 | S=90 | **S=99** | NS | NS | 1995 |
| kokarnag | - | S=90 | S=90 | **S=95** | NS | NS | 1995 |
| Qazigund | - | **S=95** | NS | **S=99** | NS | NS | 1995 |

-Decreasing; S= significance level ; NS = Insignifance;  Bold =  high significance

**Table 3: Mean temperature increase at each station from during 1980-2010.**

| Stations | Elevation in meters | Increase annual temperature in °C |
|---|---|---|
| Pahalgam | 2600mts | 1.13 |
| Gulmarg | 2740mts | 1.04 |
| Srinagar | 1600mts | 0.55 |
| Kupwara | 1670mts | 0.92 |
| Kokarnag | 2000mts | 0.99 |
| Qazigund | 1650mts | 0.78 |



| Table 4: Statistics of temperature at all the six stations of Kashmir (1980-2010) | | | | | | |
|---|---|---|---|---|---|---|
| **Gulmarg Temperature** | | Test statistic | Critical values | | | Result |
| | | | (Statistical table) | | | |
| | | | a=0.1 | a=0.05 | a=0.01 | |
| Annual | Mann-Kendall | 2.923 | 1.645 | 1.96 | 2.576 | S (0.01) |
| | Linear Regression | 3.12 | 1.699 | 2.045 | 2.756 | S (0.01) |
| | Student's t | -2.564 | 1.697 | 2.042 | 2.75 | S (0.05) |
| Maximum | | 1.782 | 1.645 | 1.96 | 2.576 | S (0.1) |
| | | 1.942 | 1.699 | 2.045 | 2.756 | S (0.1) |
| | | -2.114 | 1.697 | 2.042 | 2.75 | S (0.05) |
| Minimum | | 3.059 | 1.645 | 1.96 | 2.576 | S (0.01) |
| | | 3.79 | 1.699 | 2.045 | 2.756 | S (0.01) |
| | | -3.194 | 1.697 | 2.042 | 2.75 | S (0.01) |
| Winter | | 2.43 | 1.645 | 1.96 | 2.576 | S (0.05) |



| | | | | | |
|---|---|---|---|---|---|
| | 2.259 | 1.699 | 2.045 | 2.756 | S (0.05) |
| | -2.805 | 1.697 | 2.042 | 2.75 | S (0.01) |
| Spring | 2.006 | 1.645 | 1.96 | 2.576 | S (0.05) |
| | 2.224 | 1.699 | 2.045 | 2.756 | S (0.05) |
| | -2.025 | 1.697 | 2.042 | 2.75 | S (0.1) |
| Summer | 0.986 | 1.645 | 1.96 | 2.576 | NS |
| | 0.829 | 1.699 | 2.045 | 2.756 | NS |
| | -1.193 | 1.697 | 2.042 | 2.75 | NS |
| Autumn | 2.859 | 1.645 | 1.96 | 2.576 | S (0.01) |
| | 2.32 | 1.699 | 2.045 | 2.756 | S (0.05) |
| | -2.322 | 1.697 | 2.042 | 2.75 | S (0.05) |
| | | | | | |
| **Pahalgam Temperature** | | | | | |
| Annual | 4.119 | 1.645 | 1.96 | 2.576 | S (0.01) |
| | 3.996 | 1.645 | 1.96 | 2.576 | S (0.01) |
| | -1.985 | 1.696 | 2.04 | 2.745 | S (0.1) |
| Maximum | 3.519 | 1.645 | 1.96 | 2.576 | S (0.01) |
| | 4.457 | 1.697 | 2.042 | 2.75 | S (0.01) |
| | -3.152 | 1.696 | 2.04 | 2.745 | S (0.01) |
| Minimum | 3.6 | 1.645 | 1.96 | 2.576 | S (0.01) |
| | 4.553 | 1.697 | 2.042 | 2.75 | S (0.01) |
| | -1.554 | 1.696 | 2.04 | 2.745 | NS |
| Winter | 3.811 | 1.645 | 1.96 | 2.576 | S (0.01) |
| | 3.856 | 1.697 | 2.042 | 2.75 | S (0.01) |
| | -2.482 | 1.696 | 2.04 | 2.745 | S (0.05) |
| Spring | 3.438 | 1.645 | 1.96 | 2.576 | S (0.01) |
| | 4.597 | 1.697 | 2.042 | 2.75 | S (0.01) |
| | -3.166 | 1.696 | 2.04 | 2.745 | S (0.01) |
| Summer Temperature | 1.719 | 1.645 | 1.96 | 2.576 | S (0.1) |
| | 1.915 | 1.697 | 2.042 | 2.75 | S (0.1) |
| | -1.451 | 1.696 | 2.04 | 2.745 | NS |
| Autumn Temperature | 2.416 | 1.645 | 1.96 | 2.576 | S (0.05) |
| | 2.46 | 1.697 | 2.042 | 2.75 | S (0.05) |
| | -1.823 | 1.696 | 2.04 | 2.745 | S (0.1) |
| **Kokarnag Temperature** | | | | | |
| Annual | 3.62 | 1.645 | 1.96 | 2.576 | S (0.01) |
| | 3.998 | 1.699 | 2.045 | 2.756 | S (0.01) |
| | -2.194 | 1.697 | 2.042 | 2.75 | S (0.05) |
| Maximum | 3.11 | 1.645 | 1.96 | 2.576 | S (0.01) |




| | | | | | |
|---|---|---|---|---|---|
| | 3.622 | 1.699 | 2.045 | 2.756 | S (0.01) |
| | -3.104 | 1.697 | 2.042 | 2.75 | S (0.01) |
| Minimum | 2.763 | 1.645 | 1.96 | 2.576 | S (0.01) |
| | 2.776 | 1.699 | 2.045 | 2.756 | S (0.01) |
| | -1.28 | 1.697 | 2.042 | 2.75 | NS |
| Winter | 3.195 | 1.645 | 1.96 | 2.576 | S (0.01) |
| | 3.518 | 1.699 | 2.045 | 2.756 | S (0.01) |
| | -1.632 | 1.697 | 2.042 | 2.75 | NS |
| Spring | 3.195 | 1.645 | 1.96 | 2.576 | S (0.01) |
| | 3.469 | 1.699 | 2.045 | 2.756 | S (0.01) |
| | -2.133 | 1.697 | 2.042 | 2.75 | S (0.05) |
| Summer | 1.462 | 1.645 | 1.96 | 2.576 | NS |
| | 1.108 | 1.699 | 2.045 | 2.756 | NS |
| | -1.527 | 1.697 | 2.042 | 2.75 | NS |
| Autumn | 1.680 | 1.645 | 1.96 | 2.576 | S(0.1) |
| | 1.023 | 1.699 | 2.045 | 2.756 | NS |
| | -0.315 | 1.697 | 2.042 | 2.75 | NS |
| **Kupwara Temperature** | | | | | |
| Annual | 3.433 | 1.645 | 1.96 | 2.576 | S (0.01) |
| | 3.745 | 1.699 | 2.045 | 2.756 | S (0.01) |
| | -2.384 | 1.697 | 2.042 | 2.75 | S (0.05) |
| Maximum | 3.246 | 1.645 | 1.96 | 2.576 | S (0.01) |
| | 3.842 | 1.699 | 2.045 | 2.756 | S (0.01) |
| | -3.303 | 1.697 | 2.042 | 2.75 | S (0.01) |
| Minimum | 1.819 | 1.645 | 1.96 | 2.576 | S (0.1) |
| | 2.331 | 1.699 | 2.045 | 2.756 | S (0.05) |
| | -1.485 | 1.697 | 2.042 | 2.75 | NS |
| Winter | 1.785 | 1.645 | 1.96 | 2.576 | S (0.1) |
| | 1.797 | 1.699 | 2.045 | 2.756 | S (0.1) |
| | -1.643 | 1.697 | 2.042 | 2.75 | NS |
| Spring | 2.719 | 1.645 | 1.96 | 2.576 | S (0.01) |
| | 2.98 | 1.699 | 2.045 | 2.756 | S (0.01) |
| | -3.297 | 1.697 | 2.042 | 2.75 | S (0.01) |
| Summer | 1.785 | 1.645 | 1.96 | 2.576 | S (0.1) |
| | 2.605 | 1.699 | 2.045 | 2.756 | S (0.05) |
| | -2.477 | 1.697 | 2.042 | 2.75 | S (0.05) |
| Autumn | 2.085 | 1.645 | 1.96 | 2.576 | S (0.05) |
| | 2.003 | 1.699 | 2.045 | 2.756 | S (0.05) |
| | -1.917 | 1.697 | 2.042 | 2.75 | S (0.1) |
| | | | | | |




| Srinagar Temperature | | | | | |
|---|---|---|---|---|---|
| Annual | 2.108 | 1.645 | 1.96 | 2.576 | S (0.05) |
| | 2.243 | 1.699 | 2.045 | 2.756 | S (0.05) |
| | -2.133 | 1.697 | 2.042 | 2.75 | S (0.05) |
| Maximum | 2.804 | 1.645 | 1.96 | 2.576 | S (0.01) |
| | 3.27 | 1.699 | 2.045 | 2.756 | S (0.01) |
| | -2.456 | 1.697 | 2.042 | 2.75 | S (0.05) |
| Minimum | -1.791 | 1.645 | 1.96 | 2.576 | S(0.1) |
| | -1.799 | 1.699 | 2.045 | 2.756 | S(0.1) |
| | Infinity | 1.697 | 2.042 | 2.75 | S (0.01) |
| Winter | 1.694 | 1.645 | 1.96 | 2.576 | S(0.1) |
| | 1.871 | 1.699 | 2.045 | 2.756 | S(0.1) |
| | -0.454 | 1.697 | 2.042 | 2.75 | NS |
| Spring | 2.413 | 1.645 | 1.96 | 2.576 | S (0.05) |
| | 2.164 | 1.699 | 2.045 | 2.756 | S (0.05) |
| | -2.523 | 1.697 | 2.042 | 2.75 | S (0.05) |
| Summer | 1.374 | 1.645 | 1.96 | 2.576 | S(0.1) |
| | 1.273 | 1.699 | 2.045 | 2.756 | S(0.1) |
| | -0.174 | 1.697 | 2.042 | 2.75 | NS |
| Autumn | 0.918 | 1.645 | 1.96 | 2.576 | NS |
| | 1.099 | 1.699 | 2.045 | 2.756 | NS |
| | -0.73 | 1.697 | 2.042 | 2.75 | NS |
| Qazigund Temperature | | | | | |
| Annual | 2.057 | 1.645 | 1.96 | 2.576 | S (0.05) |
| | 1.961 | 1.645 | 1.96 | 2.576 | S (0.05) |
| | Infinity | 1.697 | 2.042 | 2.75 | S (0.01) |
| Maximum | 1.983 | 1.645 | 1.96 | 2.576 | S (0.05) |
| | 1.844 | 1.699 | 2.045 | 2.756 | S(0.05) |
| | -0.034 | 1.697 | 2.042 | 2.75 | NS |
| Minimum | 1.683 | 1.645 | 1.96 | 2.576 | S (0.1) |
| | 1.683 | 1.645 | 1.96 | 2.576 | S (0.1) |
| | Infinity | 1.697 | 2.042 | 2.75 | S (0.01) |
| Winter | 2.023 | 1.645 | 1.96 | 2.576 | S (0.05) |
| | 1.843 | 1.699 | 2.045 | 2.756 | S (0.1) |
| | -0.838 | 1.697 | 2.042 | 2.75 | NS |
| Spring | 2.036 | 1.645 | 1.96 | 2.576 | S (0.05) |
| | 1.922 | 1.699 | 2.045 | 2.756 | S (0.1) |
| | -1.996 | 1.645 | 1.96 | 2.576 | S (0.05) |
| Summer | 1.714 | 1.645 | 1.96 | 2.576 | S(0.1) |
| | -1.124 | 1.699 | 2.045 | 2.756 | NS |




|  | 0.808 | 1.697 | 2.042 | 2.75 | NS |
| Autumn | -1.802 | 1.645 | 1.96 | 2.576 | S (0.1) |
|  | -1.74 | 1.699 | 2.045 | 2.756 | S (0.1) |
|  | 1.55 | 1.697 | 2.042 | 2.75 | NS |
|  |  |  |  |  |  |

| Table 5: Statistics of precipitation at all the six stations of Kashmir (1980-2010) | | | | | | |
|---|---|---|---|---|---|---|
| **Gulmarg precipitation** | | Test statistic | Critical values | | Result | |
|  |  |  | (Statistical table) | |  |  |
|  |  | a=0.1 | a=0.05 | | a=0.01 |  |
| Annual | Mann-Kendall | -1.915 | 1.645 | 1.96 | 2.576 | S(0.1) |
|  | Linear regression | -2.442 | 1.699 | 2.045 | 2.756 | S (0.05) |
|  | Student's t | 3.214 | 1.697 | 2.042 | 2.75 | S (0.01) |
| Winter |  | -0.193 | 1.645 | 1.96 | 2.576 | S(0.1) |
|  |  | 0.186 | 1.699 | 2.045 | 2.756 | S(0.1) |
|  |  | 1.946 | 1.697 | 2.042 | 2.75 | S (0.1) |
| Spring |  | -2.515 | 1.645 | 1.96 | 2.576 | S (0.01) |
|  |  | -2.922 | 1.699 | 2.045 | 2.756 | S (0.01) |
|  |  | 3.209 | 1.697 | 2.042 | 2.75 | S (0.01) |
| Summer |  | -1.445 | 1.645 | 1.96 | 2.576 | NS |
|  |  | -0.803 | 1.699 | 2.045 | 2.756 | NS |
|  |  | -0.629 | 1.697 | 2.042 | 2.75 | NS |
| Autumn |  | -1.394 | 1.645 | 1.96 | 2.576 | NS |
|  |  | -1.428 | 1.699 | 2.045 | 2.756 | NS |
|  |  | 1.001 | 1.697 | 2.042 | 2.75 | NS |
| **Pahalgam precipitation** | |  |  |  |  |  |
| Annual |  | -0.425 | 1.645 | 1.96 | 2.576 | NS |
|  |  | -0.702 | 1.699 | 2.045 | 2.756 | S=(0.1) |
|  |  | 1.773 | 1.697 | 2.042 | 2.75 | S (0.1) |
| Winter |  | -0.176 | 1.645 | 1.96 | 2.576 | S(0.1) |
|  |  | 0.104 | 1.699 | 2.045 | 2.756 | NS |
|  |  | 1.946 | 1.697 | 2.042 | 2.75 | S (0.1) |
| Spring |  | -2.915 | 1.645 | 1.96 | 2.576 | S (0.01) |
|  |  | -2.851 | 1.699 | 2.045 | 2.756 | S (0.01) |
|  |  | 2.479 | 1.697 | 2.042 | 2.75 | S (0.05) |
| Summer |  | 1.156 | 1.645 | 1.96 | 2.576 | NS |



| | | | | | |
|---|---|---|---|---|---|
| | 1.535 | 1.699 | 2.045 | 2.756 | NS |
| | -1.387 | 1.697 | 2.042 | 2.75 | NS |
| Autumn | 0.034 | 1.645 | 1.96 | 2.576 | NS |
| | 0.348 | 1.699 | 2.045 | 2.756 | NS |
| | -0.622 | 1.697 | 2.042 | 2.75 | NS |
| **Kokarnag precipitation** | | | | | |
| *Station* Annual Precipitation | -1.326 | 1.645 | 1.96 | 2.576 | S=(0.1) |
| | -1.436 | 1.645 | 1.96 | 2.576 | S=(0.1) |
| Winter precipitation | -1.93 | 1.645 | 1.96 | 2.576 | S(0.1) |
| | -1.592 | 1.699 | 2.045 | 2.756 | NS |
| Spring Precipitation | -2.176 | 1.645 | 1.96 | 2.576 | S (0.05) |
| | -2.525 | 1.699 | 2.045 | 2.756 | S (0.05) |
| Summer Precipitation | 0.187 | 1.645 | 1.96 | 2.576 | NS |
| | -0.154 | 1.699 | 2.045 | 2.756 | NS |
| Autumn Precipitation | -0.901 | 1.645 | 1.96 | 2.576 | NS |
| | -0.903 | 1.645 | 1.96 | 2.576 | NS |
| **Srinagar precipitation** | | | | | |
| Annual | -2.532 | 1.645 | 1.96 | 2.576 | S (0.05) |
| | -2.931 | 1.699 | 2.045 | 2.756 | S (0.05) |
| | 3.094 | 1.697 | 2.042 | 2.75 | S (0.01) |
| Winter | -1.096 | 1.645 | 1.96 | 2.576 | NS |
| | -1.071 | 1.649 | 2.045 | 2.756 | NS |
| | 0.584 | 1.697 | 2.042 | 2.75 | NS |
| Spring | -2.906 | 1.645 | 1.96 | 2.576 | S (0.01) |
| | 3.741 | 1.699 | 2.045 | 2.756 | S (0.01) |
| | 3.205 | 1.697 | 2.042 | 2.75 | S (0.01) |
| Summer | -1.105 | 1.645 | 1.96 | 2.576 | NS |
| | -0.92 | 1.699 | 2.045 | 2.756 | NS |
| | 0.673 | 1.697 | 2.042 | 2.75 | NS |
| Autumn | -1.003 | 1.645 | 1.96 | 2.576 | NS |
| | -1.014 | 1.645 | 1.96 | 2.576 | NS |
| | 0.761 | 1.697 | 2.042 | 2.75 | NS |
| **Qazigund Precipitation** | | | | | |
| Annual | -2.275 | 1.645 | 1.96 | 2.576 | S(0.05) |
| | -1.976 | 1.645 | 1.96 | 2.576 | S(0.05) |
| | 1.946 | 1.697 | 2.042 | 2.75 | S (0.1) |



| | | | | | |
|---|---|---|---|---|---|
| Winter | -0.746 | 1.645 | 1.96 | 2.576 | NS |
| | -0.733 | 1.645 | 1.96 | 2.576 | NS |
| | -0.315 | 1.696 | 2.04 | 2.745 | NS |
| Spring | -2.587 | 1.645 | 1.96 | 2.576 | S(0.01) |
| | -2.706 | 1.645 | 1.96 | 2.576 | S(0.01) |
| | -0.773 | 1.696 | 2.04 | 2.745 | NS |
| Summer | 0.859 | 1.645 | 1.96 | 2.576 | NS |
| | 0.567 | 1.645 | 1.96 | 2.576 | NS |
| | -1.078 | 1.696 | 2.04 | 2.745 | NS |
| Autumn | -0.632 | 1.645 | 1.96 | 2.576 | NS |
| | -0.702 | 1.645 | 1.96 | 2.576 | NS |
| | 0.525 | 1.696 | 2.04 | 2.745 | NS |
| **Kupwara Precipitation** | | | | | |
| Annual | -1.962 | 1.645 | 1.96 | 2.576 | S (0.1) |
| | -1.059 | 1.645 | 1.96 | 2.576 | S (0.1) |
| | 3.045 | 1.699 | 2.045 | 2.756 | S (0.01) |
| Winter | -0.117 | 1.645 | 1.96 | 2.576 | NS |
| | 0.195 | 1.645 | 1.96 | 2.576 | S(0.1) |
| | 2.479 | 1.697 | 2.042 | 2.75 | S (0.05) |
| Spring | -2.962 | 1.645 | 1.96 | 2.576 | S (0.01) |
| | -3.059 | 1.645 | 1.96 | 2.576 | S (0.01) |
| | 1.773 | 1.697 | 2.042 | 2.75 | S (0.1) |
| Summer | -0.153 | 1.645 | 1.96 | 2.576 | NS |
| | -0.084 | 1.645 | 1.96 | 2.576 | NS |
| | 0.143 | 1.697 | 2.042 | 2.75 | NS |
| Autumn | -0.153 | 1.645 | 1.96 | 2.576 | NS |
| | -0.084 | 1.645 | 1.96 | 2.576 | NS |
| | -0.031 | 1.697 | 2.042 | 2.75 | NS |





**Figure captions:**

Fig. 1.Geographical setting and topographic map (elevationin meter is above mean sea level) of the
Kashmir Valley along with marked locations of six meteorological observation stations: Srinagar, Gulmarg,
Pahalgam, Kokarnag, Qazigund and Kupwara






Fig. 2 (a-g).Trends in surface temperature (°C) at the six interested locations of the Kashmir valley. (a) for
annual mean temperature, (b) maximum temperature, (c) minimum temperature, (d) winter mean
temperature during December-February, (e) spring mean temperature (March-May), (f) summer mean
temperature (June-August) and (g) autumn mean temperature (September-November).

Fig. 3 (a-e). Same as Fig. 2 but for precipitation (mm) and only for means of (a) annual, (b) winter, (c)
spring, (d) summer and (e) autumn.

Fig. 4 (a).Cumulative testing for defining change point of temperature (averaged for all the six stations of
the Kashmir valley), (b) same as (a) but for precipitation, (c) Comparison of trends of Kashmir temperature
with North Atlantic Ocean (NAO index (d) same as (c) but for precipitation, (e) regression analysis of
winter temperature and (f) regression analysis of winter precipitation.

Fig. 5. (a).Comparision between observed and WRF model simulated annually averaged temperature
(averaged for all the stations) variations for the years 1980-2010, (b) same as (a) but for spring season, (c)
for summer, (d) for autumn, (e) winter, (f) for minimum temperature and (g) maximum temperature

Fig. 6 (a-f). Observed monthly-averaged surface temperature and precipitation and ERA-interim potential
vorticities at the 350 K potential temperature and 200 mb level pressure surfaces for the station, Srinagar
during the years 1980-2010.

Fig. 7 (a-f). Same as the Fig. 6 but for Kokarnag.

Fig. 8 (a-f). Same as the Fig. 7 but for Kupwara.

Fig. 9 (a-f). Same as the Fig. 8 but for Pahalgam.

Fig. 10 (a-f). Same as the Fig. 9 but for Qazigund.

Fig. 11 (a-f). Same as the Fig. 10 but for Gulmarg.

Fig. 12 (a-f). Same as the Fig. 11 but for all the stations and during the years 2011-2016.

Fig. 13. (a-f).Synopitc scale ERA-interim meridional wind velocity covering the Jammu and Kashmir
region for sis days from 01 to 06 September 2014 (historical record flooding rainfall over this region).






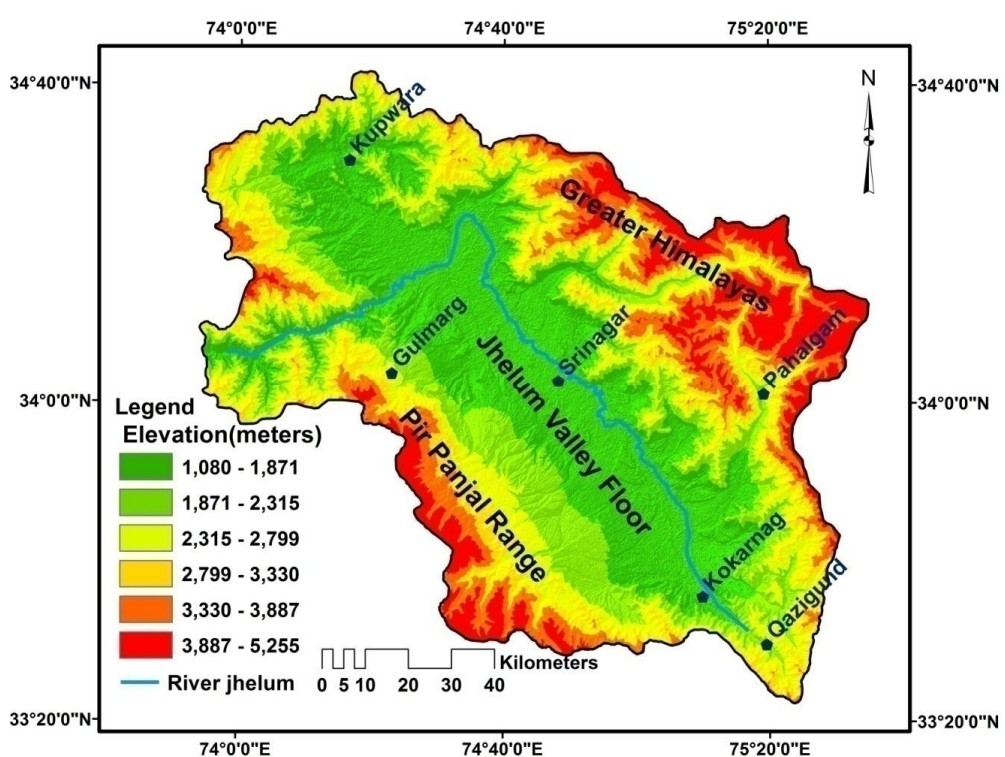



**Fig. 1**











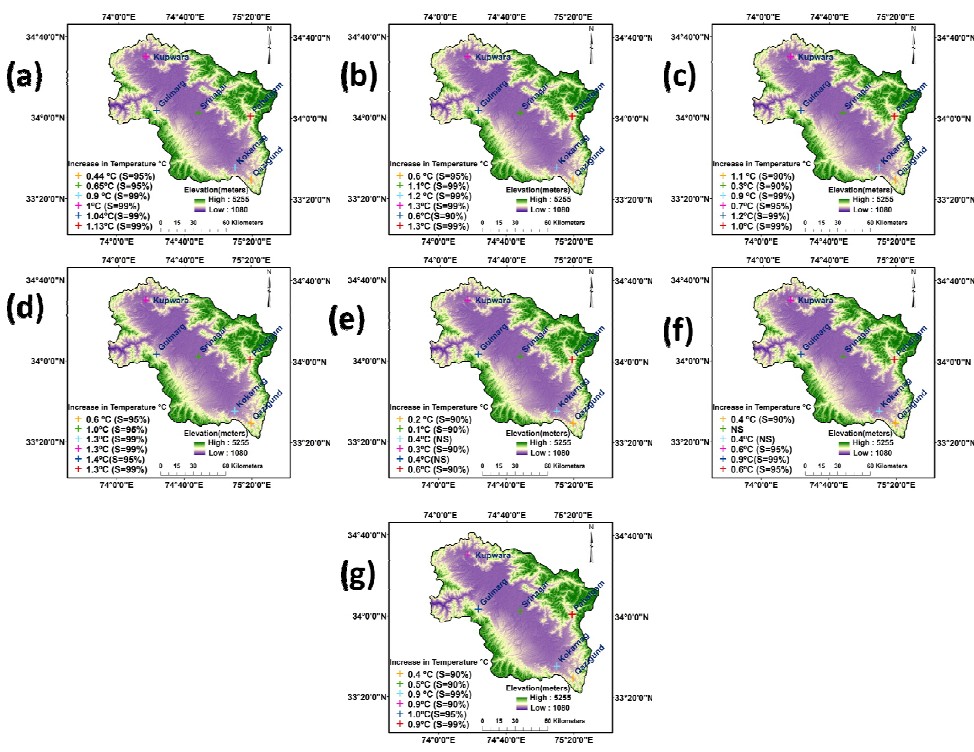



**Fig. 2**









**Fig. 3**




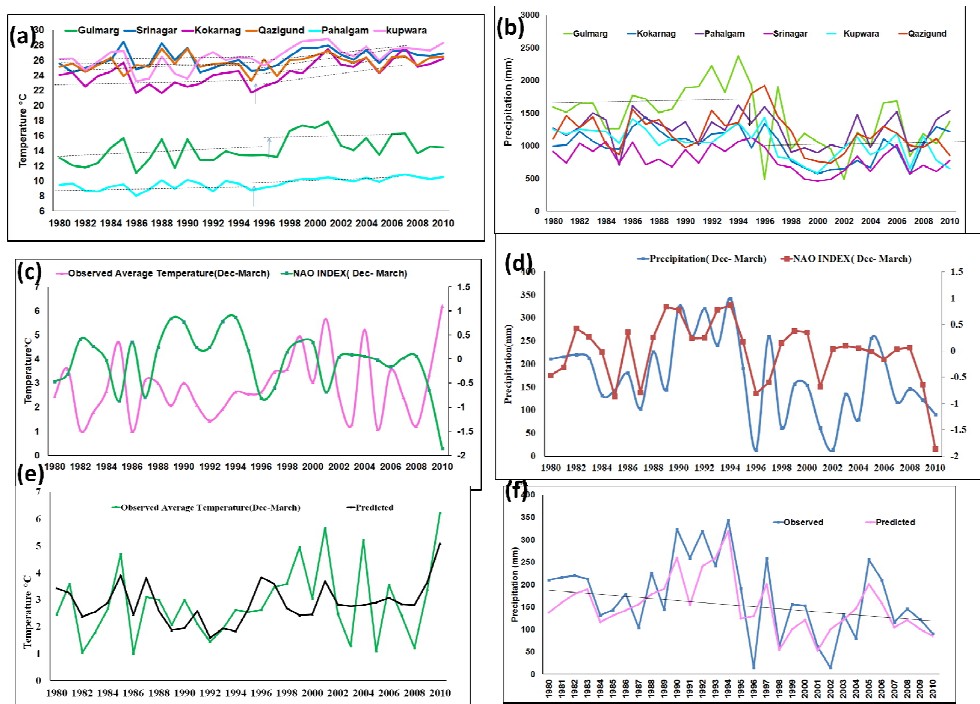

**Fig. 4**



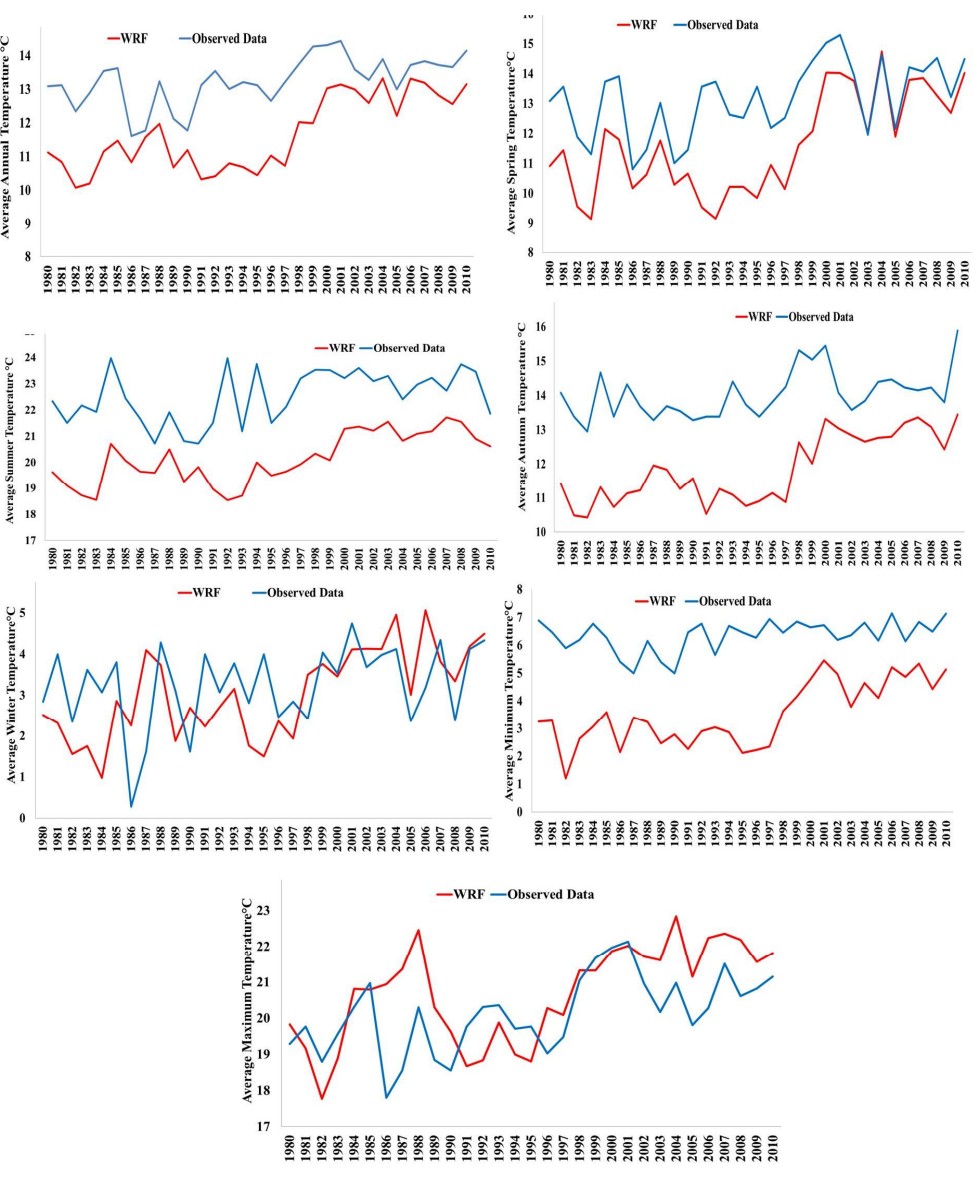


**Fig. 5**










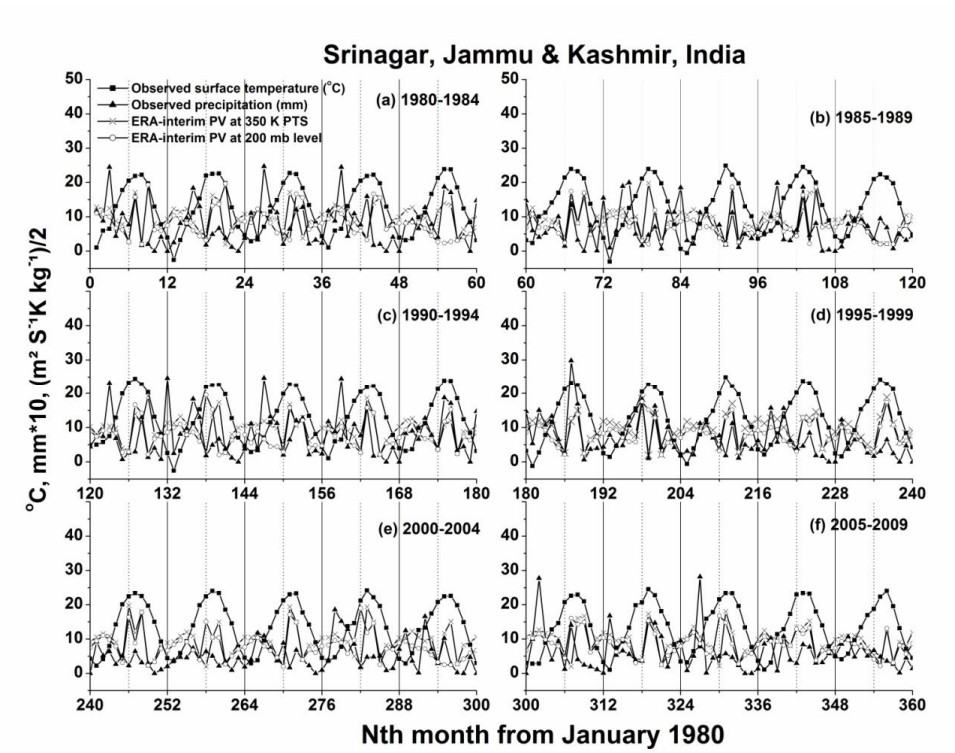


**Fig. 6**
















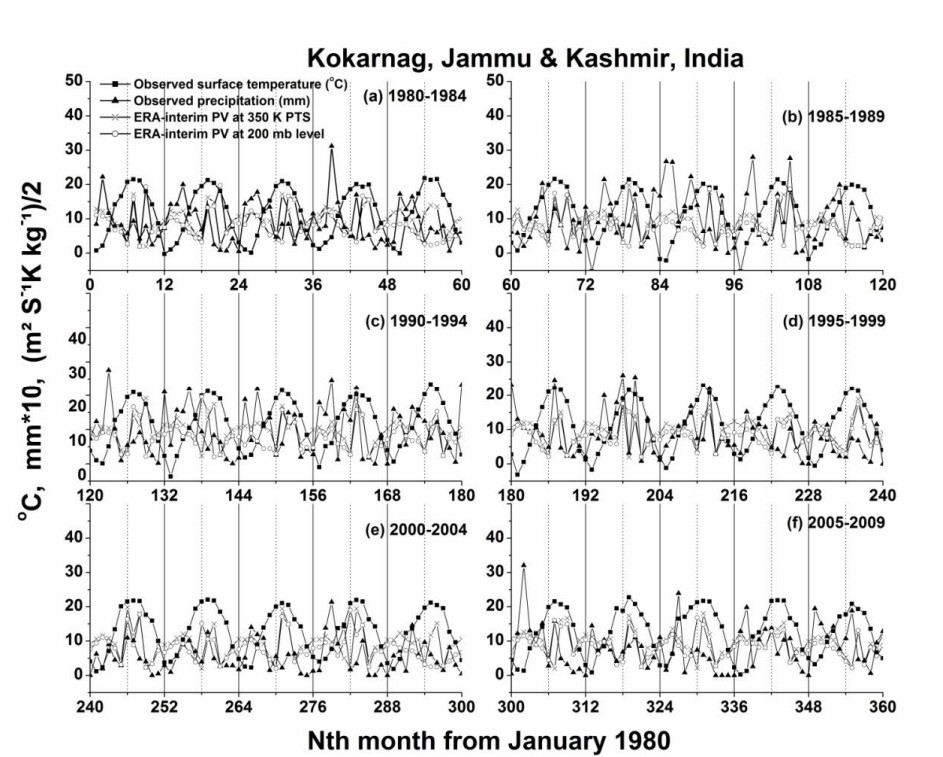


**Fig. 7**















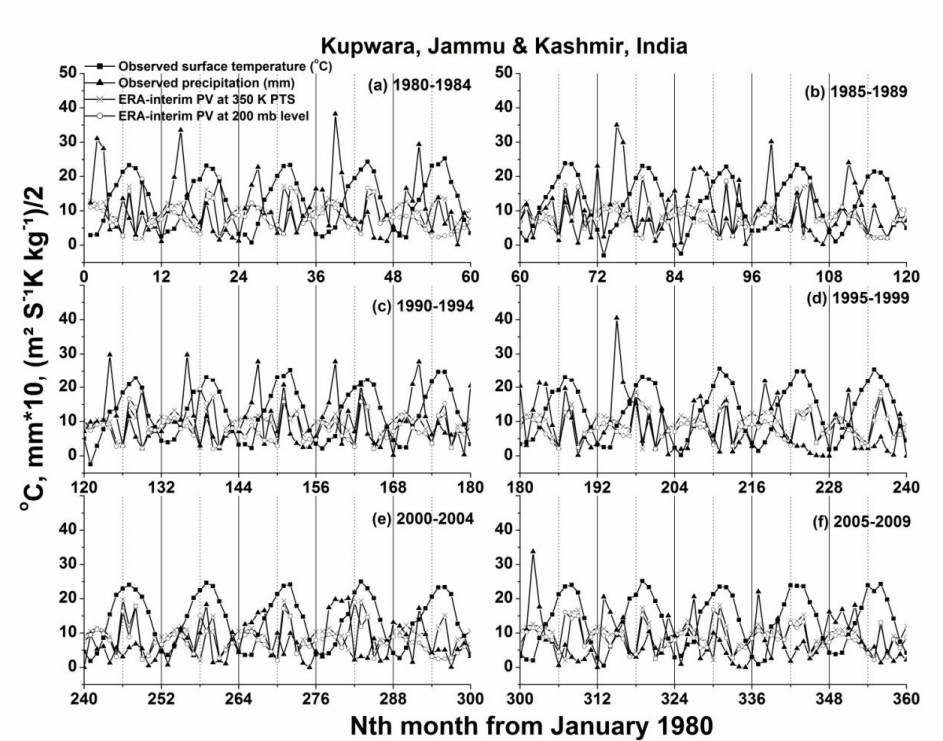


**Fig. 8**













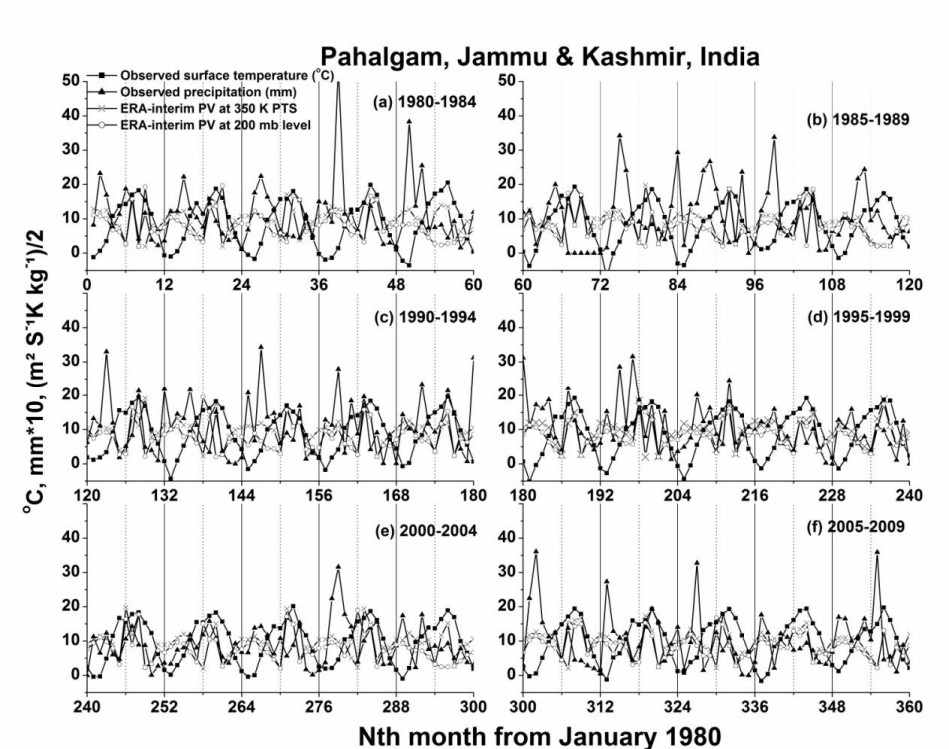

**Fig. 9**




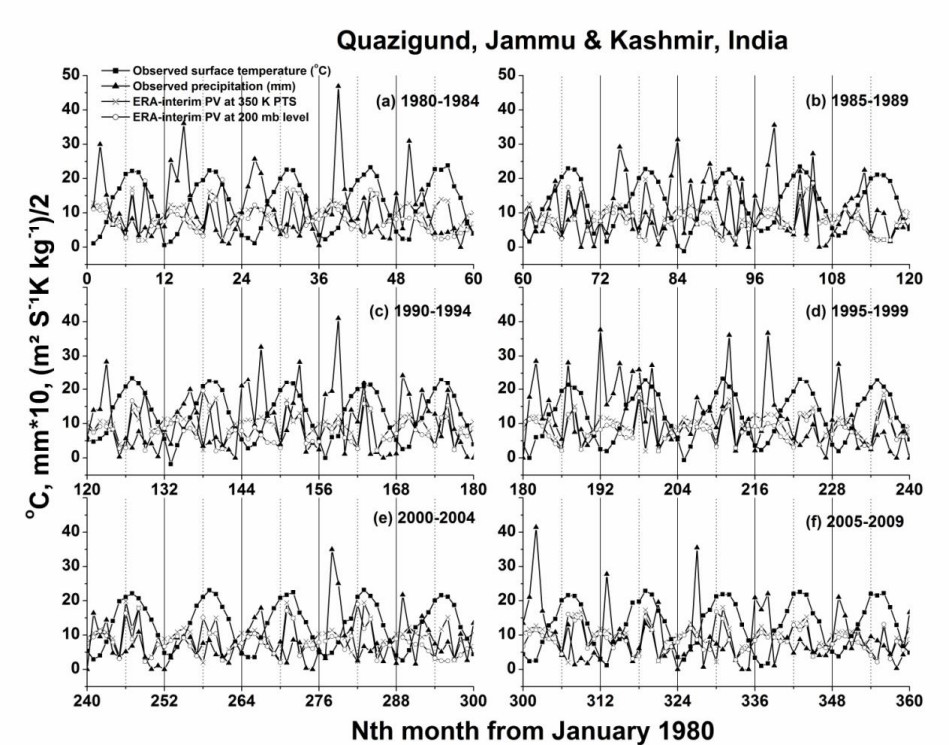


**Fig. 10**

















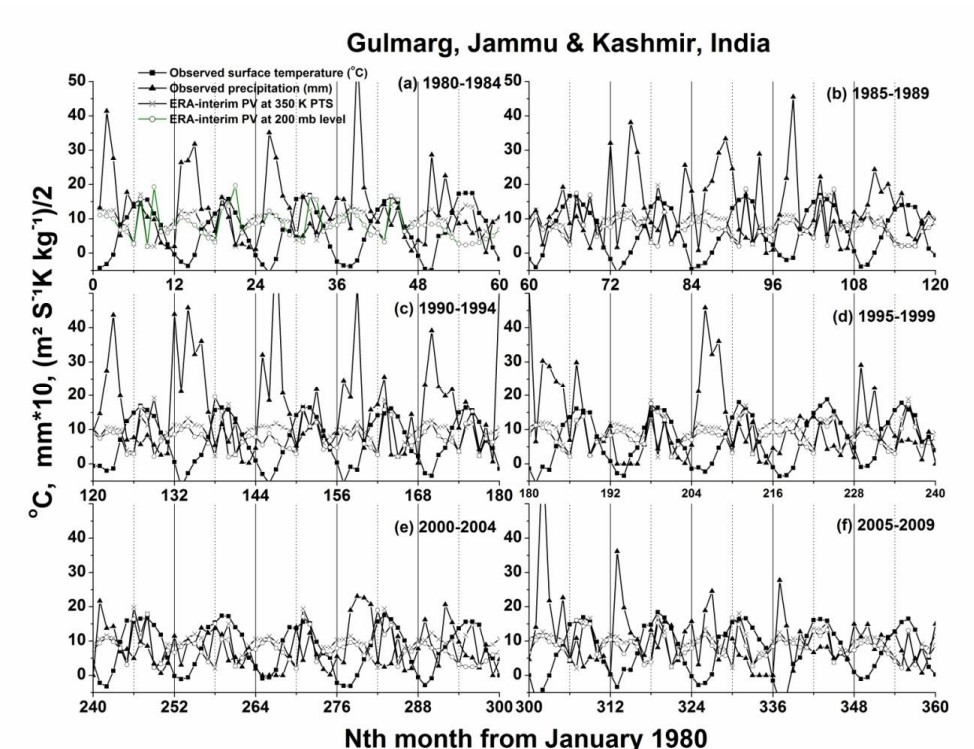


**Fig. 11**















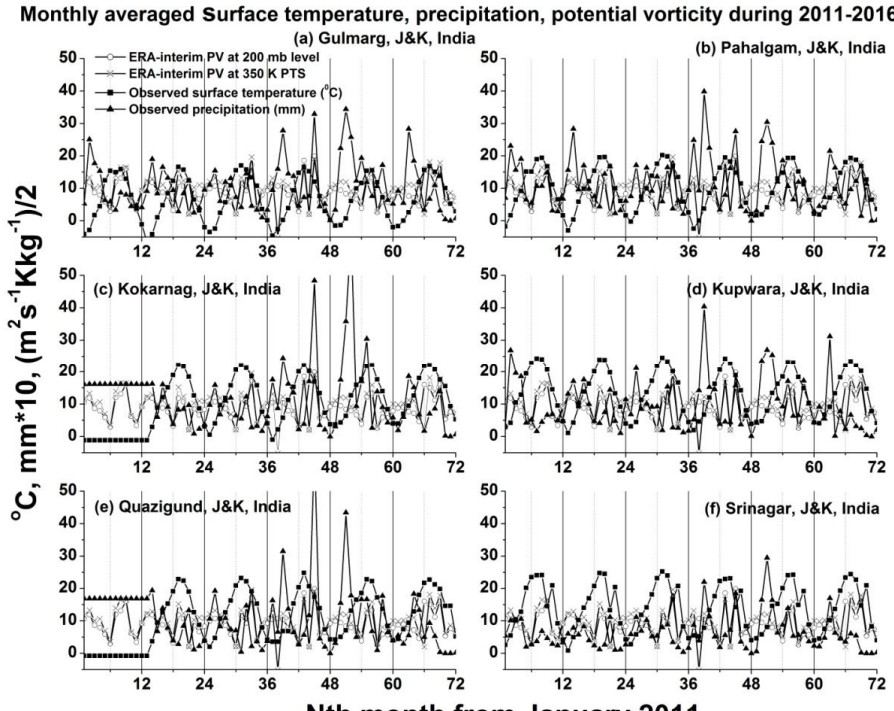


**Fig. 12**














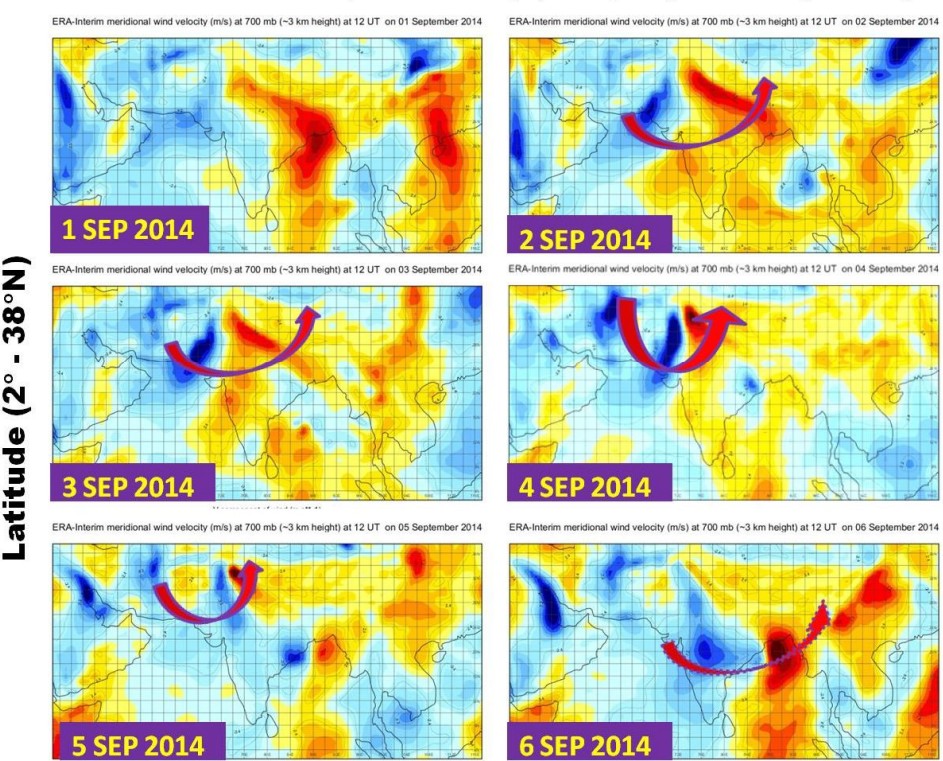


**Fig. 13**








