# Peer review of "Analyses of temperature and precipitation in the Indian Jammu-Kashmir for"

_Atmospheric Chemistry and Physics, 2018_

## Short Comment (SC1) · 11 Jun 2018

The study has been undertaken in a well defined pattern where first local temperature and precipitation has been understood and then validated and predicted with the downscaled WRF model and then finding its linkages with the local topography and global phenomena. My suggestion in this regard would be to run the WRF model at least up to 2018 so that further insight into the phenomena will be understood.

[Figure]

2018.

---

## Referee Comment (RC1) · H. Varikoden (Referee) · 3 Jul 2018

Review comments of the paper "Climatic and extreme weather variations over Mountainous Jammu and Kashmir, India: Physical explanations based on observations and modelling" by Zaz et al

This manuscript focused on climate change over the Jammu and Kashmir region in the western Himalayas. The authors present variabilities and trends of temperature and rainfall at six meteorological stations for a period of 37 years. The observed data are

compared to simulations using WRF model and are in reasonable agreement. The study also discussed the variations in temperature and precipitation with altitude and local topography. They also discussed one extreme rainfall event occurred in the first week of September 2014 and attributed that to the large-scale flow in terms of Rossby wave breaking over the region. The article is well-written at parts and highlighted the major results from the analysis. Still there are some issues to be addressed and therefore, a major revision is necessary before making it to a publication.

1. First, I am not in favor of the current title of the manuscript because the variations of extreme weather events are not addressed properly.

2. Throughout the manuscript, the authors highlight the data period of 37 years from 1980 to 2016. However, in the analysis they incorporated only from 1980 to 2010 (31 years). I suggest updating the figures and tables with updated results (1980-2016) and thus, the significance levels too. The WRF simulations are also to be updated accordingly. The NAO index is also available to date for your analyses.

3. Figure 1 can be updated with an inset figure of Jammu and Kashmir to properly identify the study region.

4. The geographical settings can be summarized in a table and delete the corresponding explanations. The table should include station name, coordinates, amsl, and remarks about the stations.

5. How the seasons are defined? The cited article did not mention anything about the seasons. Please do the classification of seasons promptly with the standard classification followed by India meteorological department or by any other classical monographs. In addition, the authors classified winter as Dec-Feb. However, in many places they considered the winter from Dec to March. This discrepancy must be corrected throughout the article. Remember, if you select the seasons differently, your interpretations and conclusions will also be affected.

6. The temporal resolution of ERA-I is missing.

7. The unit of pressure may be replaced with hPa instead of mb

8. The coordinates of second domain with 9 km resolution is not mentioned. Please update.

9. Tables 1 and 2 are not necessary, as the same information can be found in Tables 4 and 5.

10. Table 3 can be rearranged in ascending order of elevation and one more column with changes in rainfall can also be added.

11. I suggest to overly the values of changes at the respective station positions in Figures 2 and 3. The significance levels may be given in the form of a superscript star (or any other appropriate symbol) and can be indicated in the figure captions.

12. In many places, the authors quantified the changes by providing "less than" symbols. It is better to give exact values of the changes and discuss.

13. First statement in section 4.3 can be rephrased to avoid confusion.

14. Figure 4e and f show a prediction line and are in good agreement with the observed line too. Please give the corresponding regression equation for the predicted line of temperature and rainfall. Figure 4 labeling is also not correct.

15. You already discussed the skills of WRF temperature simulations in Figure 5 (please provide the station names in individual panels). In addition, the precipitation simulations can also be compared with observation to assess the performance of WRF, to complete the study.

16. The change point of temperature and rainfall is given as 1995. What is the criterion for this turning point selection? This has to be stated and substantiated with valid reasons.

17. Throughout the manuscript, the space is missing after full-stop. Also, space is missing between words in many places. Authors need to attend the typos with more care.

---

## Author Comment (AC1) · 17 Jul 2018

Responses to the reviewer (Dr. Reyaz Dar, reyazsopore@gmail.com) short comments (Interactive comment on Atmos. Chem. Phys. Discuss., https://doi.org/10.5194/acp-2018-201) of our manuscript (manuscript # acp-2018-201) titled "Climatic and extreme weather variations over Mountainous Jammu and Kashmir, India: Physical explanations based on observations and modelling" by Sumira Nazir Zaz, Romshoo Shakil Ahmad, Ramkumar Thokuluwa Krishnamoorthy, and YesuBabu Viswanadhapalli submitted for possible publication in the journal, Atmospheric Chemistry and Physics, an

open access European Geophysical Union publication

General response:

We express our sincere thanks to the reviewer for his interesting comments on our manuscript, which helps us to understand the importance of extending the data analyses up to 2018.

Specific response.

Author's query: The study has been undertaken in a well defined pattern where first local temperature and precipitation has been understood and then validated and predicted with the downscaled WRF model and then finding its linkages with the local topography and global phenomena. My suggestion in this regard would be to run the WRF model at least up to 2018 so that further insight into the phenomena will be understood.

As suggested by the reviewer, in the revised manuscript, all the results (including WRF simulations and NAO index data), figures and tables have been updated to 36 years up to December 2016 and we have not found any significant changes in trends. Because of the availability of both the observed and WRF data, at present we could update only to 2016. In the near future, we will try to update it as suggested by the reviewer.
* * *
[Figure]

**Fig. 1.** Fig. 1.Geographical setting and topographic map (elevation in meter is above mean sea level) of the Kashmir Valley along with marked locations of six meteorological observation stations: Srinagar, Gul

[Figure]

**Fig. 2.** Fig. 4 (a).Cumulative testing for defining change point of temperature (averaged for all the six stations of the Kashmir valley), (b) same as (a) but for precipitation, (c) Comparison of trends of Kas

(a)

(b)

(c)

(d)

(e)

(f)

(g)

**Fig. 3.** Fig. 5. (a). Comparision between observed and WRF model (location of Kokarnag is considered) simulated annually averaged temperature (averaged for all the stations) variations for the years 1980-2016,

**(a)**

**(b)**

**(c)**

**(d)**

**(e)**

**Fig. 4.** Fig. 6. Same as Fig. 5 but for precipitation. Here the minimum and maximum precipitation are not considered because it cannot be defined them properly in a day.

---

## Referee Comment (RC2) · Anonymous Referee #1 · 30 Jul 2018

Grammar throughout very poor, I suggest that subsequent revisions of the manuscript are proofread by a native speaker of English.

L49-50: This sentence is weirdly structured, and at any rate probably not necessary.

L58-60: These sentences need support from references: Dimri's 2015 review (doi:10.1002/2014RG000460) and Hunt et al 2018a (doi:10.1175/MWR-D-17-0258.1) would be good places to start.

L62: A reference to WD seasonality would be useful for the reader. Hunt 2018b (doi:

10.1002/qj.3200) has such a climatology, as do many studies by e.g. Dimri.

L67: You introduce the abbreviation PS here, but then don't use it again until the conclusion (where you reintroduce it anyway). I would remove it.

L72: PV is not necessarily conserved on isobaric surfaces.

L77-92: This entire paragraph has no references. Consider e.g. Rasmussen and Houze 2012 (doi: 10.1175/BAMS-D-11-00236.1), Romatschke and Houze 2011 (doi: 10.1175/2010JHM1311.1), Houze and Rasmussen 2016 (https://atmos.washington.edu/MG/PDFs/Houze-etal_Uttarakhand-Flood.pdf), Martius et al 2012 (doi:10.1002/qj.2082) and references therein.

L98: Dee et al 2011, not 2001.

You introduce the idea of WDs being important for the kind of events considered in the study, which then goes on to look at observed climate change. Why not include a short discussion on previous work that looks at the relationship between the two? E.g. Das et al 2002 (Current Science), Shekhar et al 2010 (Annals of Glaciology), Kumar et al 2015 (Atmosfera).

L186: Link incorrect.

Tables 1 and 2: This is not the correct way to perform statistical tests. You must decide on a null hypothesis, a sensible confidence value for significance testing, and then determine whether the evidence is sufficient to reject the null hypothesis at the selected confidence level. No doubt your results are significant, but you must not present them like this. If you feel the reader will benefit from these details, I will consider accepting a table of p-values, so long as the usage is clearly justified in the text.

L223-238: Lots of spaces between words seem to be missing (true throughout the manuscript but especially bad here).

Tables 4 and 5 are rather massive. I'm confident that you can substantially reduce

them in size by omitting data of low relevance, or otherwise they should be demoted to supplementary.

L273: 42 mm per what? Season?

Fig 2: Text in subfigures is too small to read. Perhaps it would be clearer to present the orography in grayscale, and then have the cross colours related to the temperature changes. Text sixe fine in Fig 3, but the other point on clarity still applies.

L291: You should state whether these correlation coefficients are significant.

L300-307: What does this paragraph (and associated figure) add to the discussion? If the purpose is to show that WRF well captures the climate features of the region, then you should state that and the implications. Personally, I think it could be removed.

L327: It is convention to refer to this as the 1997-98 El Nino

L350: Also Dimri and Dash 2012 (doi: 10.1007/s10584-011-0201-y)

L367: You say the rate of warming has been especially high in the last forty years and then have two references to twenty-year-old papers.

Fig 4: Why are some of the data smoothed and others not?

L406: And Lau and Kim 2012 (doi: 10.1175/JHM-D-11-016.1)

The first and third paragraphs of Sec 4.5 (though you have it labelled as 4.3) might be better placed in the introduction, but I will leave this up to the authors.

Fig 6: Difficult to tell the lines apart, could you use colour? At present, it is difficult to distil any useful information from this. This analysis could benefit from including discussion of western disturbance frequency, if possible.

L466: "due to the effect of climate change" – how have you deduced this?

Fig 13: 700 hPa is quite a low level to be looking at Rossby wave activity, what is the structure at higher levels?

L446-532: I feel that the text in this section is overly explicit, and could be easily reduced for improved readability.

---

## Author Response (AR1)

Responses to the reviewer comments of our manuscript (manuscript # acp-2018-201) titled "*Climatic and extreme weather variations over Mountainous Jammu and Kashmir, India: Physical explanations based on observations and modelling*" by

Sumira Nazir Zaz, Romshoo Shakil Ahmad, Ramkumar Thokuluwa Krishnamoorthy, and YesuBabu Viswanadhapalli submitted for possible publication in the journal, *Atmospheric Chemistry and Physics,* an open access European Geophysical Union publication.

**General response:**

We express our sincere thanks to the reviewer for his invaluable and insightful comments on our manuscript, which helps us to improve the quality of it immensely. Below we have provided our one to one responses to his all queries and we hope that the reviewer will be convinced of our responses and make positive recommendations to this revised manuscript, written by taking into account of his as well as the other reviewers' comments (Prof. Reyaz Dar and Dr. H. Varikoden, responses to their comments are pasted below, please find), for its possible publication in the journal, *Atmospheric Chemistry and Physics,* an open access European Geophysical Union publication. We have also enclosed the revised manuscript with changes tracked version. Our responses following each of the comments (in red color) are marked in blue color

**One to one response to the reviewer comments:**

L49-50: This sentence is weirdly structured, and at any rate probably not necessary

The sentence has been corrected

L58-60: These sentences need support from references: Dimri's 2015 review (doi:10.1002/2014RG000460) and Hunt et al 2018a (doi:10.1175/MWR-D-17-0258.1) would be good places to start.

These references are now incorporated.

L62: A reference to WD seasonality would be useful for the reader. Hunt 2018b (doi: C1 ACPD Interactive comment Printer-friendly version Discussion paper 10.1002/qj.3200) has such a climatology, as do many studies by e.g. Dimri

These references are now added

L67: You introduce the abbreviation PS here, but then don't use it again until the conclusion (where you reintroduce it anyway). I would remove it.

The abbreviation PS has been removed

L72: PV is not necessarily conserved on isobaric surfaces.

Potential vorticity on isobaric surfaces is considered as a conserved quantity in the case of Rossby waves generated due to large scale wind flows over topography. In this case, the waves are called barotrophic Rossby waves.

L77-92: This entire paragraph has no references. Consider e.g. Rasmussen and Houze 2012 (doi: 10.1175/BAMS-D-11-00236.1), Romatschke and Houze 2011 (doi: 10.1175/2010JHM1311.1), Houze and Rasmussen 2016 (https://atmos.washington.edu/MG/PDFs/Houze-etal_Uttarakhand-Flood.pdf), Martius et al 2012 (doi:10.1002/qj.2082) and references therein.

These references have been incorporated in the paragraph.

L98: Dee et al 2011, not 2001.

Corrected the reference

L186: Link incorrect.

 It is now corrected.

Tables 1 and 2: This is not the correct way to perform statistical tests. You must decide on a null hypothesis, a sensible confidence value for significance testing, and then determine whether the evidence is sufficient to reject the null hypothesis at the selected confidence level. No doubt your results are significant, but you must not present them like this. If you feel the reader will benefit from these details, I will consider accepting a table of p-values, so long as the usage is clearly justified in the text.

The suggestion has been incorporated.

L223-238: Lots of spaces between words seem to be missing (true throughout the manuscript but especially bad here).

It is now corrected

Tables 4 and 5 are rather massive. I'm confident that you can substantially reduce them in size by omitting data of low relevance, or otherwise they should be demoted to supplementary.

The tables 4 and 5 omitted and the information in the table is now added in the tables 1 and 2. This is suggested also by other reviewer.

L273: 42 mm per what? Season?

42 mm per spring seasons during these 37 years, which is mentioned in the text now.

Fig 2: Text in subfigures is too small to read. Perhaps it would be clearer to present the orography in grayscale, and then have the cross colours related to the temperature changes. Text sixe fine in Fig 3, but the other point on clarity still applies.

The suggestion has been incorporated in Fig 2 and Fig3.

L291: You should state whether these correlation coefficients are significant.

The significance has been mentioned

L300-307: What does this paragraph (and associated figure) add to the discussion? If the purpose is to show that WRF well captures the climate features of the region, then you should state that and the implications. Personally, I think it could be removed.

It is stated now and the implications

L327: It is convention to refer to this as the 1997-98 El Nino

Now it is referred as 1997-98 El Nino

L350: Also Dimri and Dash 2012 (doi: 10.1007/s10584-011-0201-y)

Reference has been included.

L367: You say the rate of warming has been especially high in the last forty years and then have two references to twenty-year-old papers.

Some new references have been added now.

Fig 4: Why are some of the data smoothed and others not?

All the data have been smoothed

L406: And Lau and Kim 2012 (doi: 10.1175/JHM-D-11-016.1)

Added

The first and third paragraphs of Sec 4.5 (though you have it labelled as 4.3) might be better placed in the introduction, but I will leave this up to the authors

We fear that continuity of information will be lost if we put these two paragraphs in the introduction section

Fig 6: Difficult to tell the lines apart, could you use colour? At present, it is difficult to distil any useful information from this. This analysis could benefit from including discussion of western disturbance frequency, if possible

Now it is colored.

L466: "due to the effect of climate change" – how have you deduced this?

Modified these words to avoid ambiguities

Fig 13: 700 hPa is quite a low level to be looking at Rossby wave activity, what is the structure at higher levels?

Baroclinic Rossby waves are manifested simultaneously well in the lower and upper tropospheres. We afraid that number of figures will be increased further if we add information at higher levels as there are already 14 figures.

L446-532: I feel that the text in this section is overly explicit, and could be easily reduced for improved readability

Now it is improved for readability

Responses to the reviewer (Dr. H. Varikoden, hamza@tropmet.res.in) comments (interactive comment on Atmos. Chem. Phys. Discuss., https://doi.org/10.5194/acp-2018-201, 2018) of our manuscript (manuscript # acp-2018-201) titled "*Climatic and extreme weather variations over Mountainous Jammu and Kashmir, India: Physical explanations based on observations and modelling*" by Sumira Nazir Zaz, Romshoo Shakil Ahmad, Ramkumar Thokuluwa Krishnamoorthy, and YesuBabu Viswanadhapalli submitted for possible publication in the journal, *Atmospheric Chemistry and Physics,* an open access European Geophysical Union publication.

**General response:**

We express our sincere thanks to the reviewer for his invaluable comments on our manuscript, which helps us to improve the quality of it immensely. Below we have provided our one to one responses to his all queries and we hope that the reviewer will be convinced of our responses and make positive recommendations to this revised manuscript, written by taking into account of his as well as the other reviewer's short comment (Prof. Reyaz Dar), for its possible publication in the journal, *Atmospheric Chemistry and Physics,* an open access European Geophysical Union publication. We have also enclosed the revised manuscript with changes tracked version. Our responses following each of the comments are marked as **bold faceted words.**

**One to one response to the reviewer comments:**

1. First, I am not in favour of the current title of the manuscript because the variations of extreme weather events are not addressed properly.

   Now the title is modified as
   **Climate and the September 2014 flood event over Mountainous Jammu and Kashmir, India: Physical explanations based on observations and modelling**

2. Throughout the manuscript, the authors highlight the data period of 37 years from 1980 to 2016. However, in the analysis they incorporated only from 1980 to 2010 (31 years). I suggest updating the figures and tables with updated results (1980-2016) and thus, the significance levels too. The WRF simulations are also to be updated accordingly. The NAO index is also available to date for your analyses.

   **In the revised manuscript, all the results (including WRF simulations and NAO index data), figures and tables have been updated to 36 years up to 2016 and we have not found any significant changes in trends.**

3. Figure 1 can be updated with an inset figure of Jammu and Kashmir to properly identify the study region

   **Figure 1 has been updated.**

4. The geographical settings can be summarized in a table and delete the corresponding explanations. The table should include station name, coordinates, amsl, and remarks about the stations.

**Corrections as suggested by the reviewer have been incorporated.**

5. How the seasons are defined? The cited article did not mention anything about the seasons. Please do the classification of seasons promptly with the standard classification followed by India meteorological department or by any other classical monographs. In addition, the authors classified winter as Dec-Feb. However, in many places they considered the winter from Dec to March. This discrepancy must be corrected throughout the article. Remember, if you select the seasons differently, your interpretations and conclusions will also be affected.

**The reviewer can get clarified that with respect to NAO (Fig. 4) only it is considered December-March as winter months and in all other parts of the manuscript, December-February is considered as winter season as per the IMD definition. This is because, for the NAO index, normally December-March is considered as northern winter and we adopted the same definition here (Archer and Fowler, 2004; Iqbal and Kashif,2013). Since the result of linkage between winter NAO index and Kashmir precipitation does not affect other results of this manuscript, we don't need to do any corrections for other places. This explanation is provided in the revised manuscript while discussing the Fig. 4 as well as in the section 2.**

6. The temporal resolution of ERA-I is missing.
**The temporal resolution of ERA-I is monthly averaged, which is now mentioned in the revised manuscript.**

7. The unit of pressure may be replaced with hPa instead of mb
   **mb is now replaced by hPa.**

8. The coordinates of second domain with 9 km resolution is not mentioned. Please update.

**It is now updated.**

**The dimensions of the WRF model domains are listed below**

**Domain -1 (18-km ) extends from  Longitude from 24.8516 E to 115.148E and Latitudes from 22.1127S to 46.7629 N**

**Domain -2 (6-km ) extends from Longitude from 56.3838E to 98.5722E and Latitude from 3.86047 S to 38.2874 N**

9. Tables 1 and 2 are not necessary, as the same information can be found in Tables 4 and 5.

**This suggestion is well taken and has been incorporated**

10. Table 3 can be rearranged in ascending order of elevation and one more column with changes in rainfall can also be added.

**The table is arranged in descending order to show stations with higher increase in temperature. One more column of topography has been incorporated.**

11. I suggest to overly the values of changes at the respective station positions in Figures 2 and 3. The significance levels may be given in the form of a superscript star (or any other appropriate symbol) and can be indicated in the figure captions.

**We are sorry that it is difficult now to do and we will surely attempt to do as suggested by the reviewer during the final phase of publication if it is recommended for.**

12. In many places, the authors quantified the changes by providing "less than" symbols. It is better to give exact values of the changes and discuss.

**Exact values at corresponding confidence levels are already provided in closed brackets. The reviewer may kindly look for it in the manuscript. The following italicised sentences are there already in the starting portion of the section 4.**

**"S in S=99%" indicates statistically significant. It is to be noted that hereafter it will not be mentioned explicitly about the period 1980-2016 and the statistical significance of derived values. All the results are subjected to statistical tests with confidence level of statistical significance at S=99% unless otherwise mentioned explicitly. Further, values denoting "less than" refer to S=99% and the confidence levels corresponding to given values are provided within closed brackets.**

13. First statement in section 4.3 can be rephrased to avoid confusion.
**The statement has been now rephrased**

14. Figure4e and f show a prediction line and are in good agreement with the observed line too. Please give the corresponding regression equation for the predicted line of temperature and rainfall.

**The following algorithm from Microsoft Excel defines the forecast method applied here and this is now included in the revised manuscript.**
**The forecast algorithm calculates or predicts a future value by using existing values. The predicted value is a y-value for a given x-value. The known values are existing x-values and y-values, and the new value is predicted by using linear regression.**

**The syntax is as follows**

**FORECAST(x, known_y's, known_x's)**

**X** is the data point for which we want to predict a value.

**Known_y's** is the dependent array or range of data.
**Known_x's** is the independent array or range of data.

**The equation for FORECAST is a+bx, where:**

$$a = \bar{y} - b\bar{x} \qquad \text{and} \qquad b = \frac{\sum(x-\bar{x})(y-\bar{y})}{\sum(x-\bar{x})^2}$$

**and where x and y are the sample means AVERAGE(known_x's) and**

**AVERAGE(known y's**

Figure 4 labelling is also not correct.
**It is now corrected**

15. You already discussed the skills of WRF temperature simulations in Figure 5 (please provide the station names in individual panels).

**The reference simulations of WRF model is for the Kokarnag station which is now mentioned in the revised manuscript.**

16. In addition, the precipitation simulations can also be compared with observation to assess the performance of WRF, to complete the study.

**It is now compared the WRF precipitation data also (Fig. 6 now) and the results are discussed.**

17. The change point of temperature and rainfall is given as 1995. What is the criterion for this turning point selection? This has to be stated and substantiated with valid reasons.

**The turning point in temperature and precipitation as already mentioned in the manuscript is calculated using the Cumulative Deviation (parametric test for step jump in mean), however to validate the results distribution-Free CUSUM (non-parametric test for step jump in mean) has also been used in this study. We followed this reference in this regard.**

*Ahmad Reza Ghasemi,Changes and trends in maximum, minimum and mean temperature series in Iran,* **DOI: 10.1002/asl2.569 ,** *Atmos. Sci. Let.* **16: 366–372 (2015).**

Responses to the reviewer (Dr. Reyaz Dar**,** reyazsopore@gmail.com) short comments (Interactive comment on Atmos. Chem. Phys. Discuss., https://doi.org/10.5194/acp-2018-201) of our manuscript (manuscript # acp-2018-201) titled "*Climatic and extreme weather variations over Mountainous Jammu and Kashmir, India: Physical explanations based on observations and modelling*" *by* Sumira Nazir Zaz, Romshoo Shakil Ahmad, Ramkumar Thokuluwa Krishnamoorthy, and YesuBabu Viswanadhapalli submitted for possible publication in the journal, *Atmospheric Chemistry and Physics,* an open access European Geophysical Union publication

**General response:**

**We express our sincere thanks to the reviewer for his interesting comments on our manuscript, which helps us to understand the importance of extending the data analyses up to 2018.**

**Specific response.**

*Author's query*: The study has been undertaken in a well defined pattern where first local temperature and precipitation has been understood and then validated and predicted with the downscaled WRF model and then finding its linkages with the local topography and global phenomena. My suggestion in this regard would be to run the WRF model at least up to 2018 so that further insight into the phenomena will be understood.

**As suggested by the reviewer, in the revised manuscript, all the results (including WRF simulations and NAO index data), figures and tables have been updated to 36 years up to December 2016 and we have not found any significant changes in trends. Because of the availability of both the observed and WRF data, at present we could update only to 2016. In the near future, we will try to update it as suggested by the reviewer**

[revised manuscript text omitted]
, December-February is considered as winter season as per the IMD definition. This is because, for the NAO index, normally December-March is considered as northern winter and we adopted the same definition here(Archer and Fowler, 2004;Iqbal and Kashif,2013). The result of linkage between winter NAO index and Kashmir precipitation does not affect other results of the present work.~~ It is to be clarified that with respect to NAO (Fig. 4) only, it is considered December-March as winter months and in all other parts of the manuscript, December-February is considered as winter season as per the IMD definition. This is because, for the NAO index, normally December-March is considered as northern winter and we adopted the same definition here(Archer and Fowler, 2004;Iqbal and Kashif,2013). The result of linkage between winter NAO index and Kashmir precipitation does not affect other results of the present work. The annual temperature in the valley varies from about -10°C to 35°C. The rainfall pattern in the valley is dominated by winter time precipitation associated with western disturbances (Dar et al., 2014) while the snow precipitation is received mainly in winter and early spring season (Kaul and Qadri, 1979).

**3. Data and Methodology**

India Meteorological Department (IMD) provided 37 years (1980-2016) of data of daily precipitation, maximum temperature and minimum temperaturefor all these six stations. Monthly averaged data were further analysed to find long term variations of the weather parameters. Statistical tests including Mann-Kendall, Spearman Rho, Cumulative deviation, Student's t-test were performed to determine long term trends and turning point of weather parameters with statistical significances. Similar analyses and tests were performed also for the Weather and Research Forecasting (WRF) model simulated and ERA-Interim reanalyses data (0.75° by 0.75° spatial resolution in the horizontal plane, monthly averaged time resolution) of same weather parameters and for the NAO index. Brief information about these datasets is provided below.

**3.1.Observational and model datasets used in this study**

The obtained observational data were carefully analysed for homogeneity and missing values. Analyses of ratios of temperature from the neighbouring stations with the Srinagar station were conducted using relative homogeneity test (WMO, 1970). It is found that there is no significant inhomogeneity and data gap for any station. Few missing data points were linearly interpolated and enough care was taken not to make any meaningful interpretation during such short periods of data gap in the observations.Annual and seasonal means of temperature and precipitation were calculated for all the stations and years. To compute seasonal means, the data were divided into the following seasons: winter (December to February), spring (March to May), summer (June to August) and autumn (September to November).Trends in the annual and seasonal means of temperature and precipitationwere determined usingMann–Kendall (non-parametric test) and linear regression tests (parametric test)at the confidence

levelsof S=99% or (0.01), S=95% or 0.05 and S=90% or 0.1. These tests have been extensively used in hydrometeorological data analysesas theyare less sensitive to heterogeneity of data distribution and least affected by extreme values or outliers in data series.Various methods have been applied to determine change points of a time series (Radziejewski et al., 2000; Chen and Gupta, 2012). In this study, change point in time series of temperature and precipitationwas identified using cumulative deviation test and student's t test (Pettitt, 1979). This method detects the time of significant change in the mean of a time series when the exact time of the change is unknown (Gao et al., 2011).

The data of winter NAO index during 1980–2010from Climatic Research Unitwereobtained for analyses from the web link https// www.cru.uea.ac.uk/datahtpp://cru@uea.ac.uk. The winter (December - March) NAO index is based on the difference of normalized sea level pressure (SLP) between Lisbon, Portugal and Iceland, which is available from 1964 onwards. Positive NAO index isassociated with stronger-than-average westerlies over the middle latitudes (Hurrell,1997).  Correlation between climatic variations of mean (December-March) temperature, precipitationand NAO index was determined using Pearson correlation coefficient method.To test whether the observed trends in winter temperature and precipitation are enforced by NAO, linear regression analysis (forecast) was performed (Fig. 4e and f).The following algorithm from Microsoft Excel defines the forecast method applied here. The forecast algorithm calculates or predicts a future value by using existing values. The predicted value is a y-value for a given x-value. The known values are existing x-values and y-values, and the new value is predicted by using linear regression.

The syntax is as follows

FORECAST(x, known_y's, known_x's)

X  is the data point for which we want to predict a value.
Known_y's  is the dependent array or range of data (rainfall or temperature).
Known_x's  is the independent array or range of data (time).

The equation for FORECAST is a+bx, where:

$$a = \bar{y} - b\bar{x}$$ and

$$b = \frac{\sum (x - \bar{x})(y - \bar{y})}{\sum (x - \bar{x})^2}$$

and where x and y are the sample means AVERAGE(known_x's) and AVERAGE(known y's).

**3.2. WRF Model configuration**

The Advanced Research WRF version 3.9.1 model simulation wasused in this study to downscale theERA-Interim (European Centre for Medium Range Weather Forecasting ReAnalysis) data over the Indian Monsoon region. The model is configured with 2two-way nested domains(18 km and 9-km horizontal resolutions), 51 vertical levels and model top at 10 hPa level.The first domain of the model extends over the whole Indian monsoon region (28°E to

Domain -1 (18-km ) extends from Longitude from 24.8516 E to 115.148E and Latitudes from 22.1127S to 46.7629 N

Domain -2 (6-km ) extends from Longitude from 56.3838E to 98.5722E and Latitude from 3.86047 S to 38.2874 N

[revised manuscript text omitted]

During the summer months, precipitation follows the same decreasing trend but not at significant level (NS) forGulmarg, Kupwara, Kokarnag, Pahalgam and Srinagar (Fig. 3d, and Table 2). In addition, Qazigundshows no trendin summer precipitation. The autumn precipitation at Pahalgam,Kupwara, Kokarnag, Srinagar, Gulmarg and Qazigund shows decrease at insignificant level (NS) (Fig. 3e and Table 2).Cumulative test was used to determine the change point oftrend in the annual and seasonal variations oftemperature and precipitation. The results reveal that the year 1995 is identified as the year of abrupt increase (change point) in temperature of the valley (Fig. 4a) and the sameis identified as the year of abrupt decrease for precipitation (Fig. 4b).

**4.3. North Atlantic Oscillation (NAO) index and winter climatic fluctuations**

 tThe present study also investigated the teleconnection between theactivity of(NAO index) with and temperature and precipitation over the Kashmir valley, particularly  during the northern winter season (December - March). The results indicated that NAO hassignificant (S=0.05) negativecorrelations(-0.54) with  winter precipitation while the winter temperature showssignificant (0.01) positive correlation (0.68)  with NAO  (Fig. 4c). This  suggests that winter precipitation and temperature has some association with the winter NAO index. Hhigher precipitation over Kashmiris associated with positive phase of NAO.

with more precipitation.Further the abrupt change in Interestingly, similar to the observed temperature and precipitation from infrom 1995changes in Kashmir in mid-nineties,has clear shows association with abrupt variation in the NAO index from the same year.
[revised manuscript text omitted]

**Hunt, K., Turner, A.** and **Shaffrey, L.** (2018b) *The evolution, seasonality, and impacts of western disturbances.* Quarterly Journal of the Royal Meteorological Society, 144 (710). pp. 278-290. ISSN 1477-870X doi: **https://doi.org/10.1002/qj.3200**

Dimri AP, Niyogi D, Barros AP, Ridley J, Mohanty UC, Yasunari T and Sikka DR (2015) Western Disturbances: A Review. Reviews of Geophysics, 2014RG000460.Doi: 10.1002/2014RG000460.

Hunt K. M. R.  Turner A. G.   ShaffreyL C. (2018b) The evolution, seasonality and impacts of western disturbances,Volume144, Issue710, January 2018 278-29Part 11 November 2017 https://doi.org/10.1002/qj.3200

Romatschke, U., and R. Houze (2011), Characteristics of Precipitating Convective Systems in the Premonsoon Season of South Asia, J. Hydrometeorology, 12, 157-180, doi:10.1175/2010JHM1311.1

Rasmussen, K. L. R., and R. Houze (2012), A Flash-Flooding Storm At The Steep Edge Of High Terrain: Disaster in the Himalayas, Bull. Am. Meteorol. Soc., 93, 1713-1724, doi:10.1175/BAMS-D-11-00236.1.

Martius O, Sodemann H, Joos H, Pfahl S, Winschall A, Croci-Maspoli M, Graf M, Madonna E, Mueller B, Schemm S, Sedla´cek J, Sprenger M, Wernli H. (2012). The role of upper-level dynamics and surface processes for the Pakistan flood of July 2010. Q. J. R. Meteorol. Soc. 139: 1780–1797, doi:10.1002/qj.2082.

Das ,M.R ., Mukhopadhyay,R.L., Dandekar,M.M and Kshirsagar ,S.R,(2002). Pre-monsoon western disturbances in relation to monsoon rainfall, its advancement over NW India and their trends, current science 82(11):1320-1321

Shekhar,M.S., Chand,H., kumar,S., Ganju,Ashwagosh(2010) Climate change studies in western Himalaya, Annals of Glaciology 51(54):105-112

Kumar N, Yadav BP, Gahlot S and Singh M (2015) Winter frequency of western disturbances and precipitation indices over Himachal Pradesh, India: 1977□2007. Atmósfera 28(1), 63□70.Doi: http://dx.doi.org/10.1016/S0187□6236(15)72160□0.

Dimri,A and Dash.S., 2012. "Wintertime climatic trends in the western Himalayas," Climatic Change, Springer, vol. 111(3), pages 775-800

Hijioka Y, Lin E, Pereira JJ, Corlett RT, Cui X, Insarov GE, Lasco RD, Lindgren E and Surjan A (2014) Asia. In: Barros VR et al. (eds),Climate Change 2014: Impacts, Adaptation, and Vulnerability. Part B: Regional Aspects. Contribution of Working Group II

to the Fifth Assessment Report of the Intergovernmental Panel of Climate Change. Cambridge University Press,Cambridge, United Kingdom and New York, NY, USA

Madhura RK, Krishnan R, Revadekar JV, Mujumdar M and Goswami BN (2015) Changes in western disturbances over the Western Himalayas in a warming environment.Climate Dynamics 44(3□4), 1157□1168.Doi: 10.1007/s00382□014□2166□9.

Bookhagen B (2010) Appearance of extreme monsoonal rainfall events and their impact on erosion in the Himalaya.Geomatics,Natural Hazards and Risk 1(1), 37□50. Doi: 10.1080/19475701003625737.

Peters GP, Andrew RM, Boden T, Canadell JG, Ciais P, Le Quéré C, et al. (2013). The challenge to keep global warming below 2°C. Nature Climate Change.;3(1):4–6.

Knutti R, Rogelj J, Sedláček J, Fischer EM. A scientific critique of the two-degree climate change target. (2016). Nature Geoscience.;9(1):13–18

Lau WKM and Kim K-M(2012). The 2010pakistan flood and Russian heat wave:Teleconnection of hydrometeorological Extremes, Journal of Hydrometeorological 13(1):392-403 DOI:10.1175/jhm-D-11-016.1

[revised manuscript text omitted]

---

## Referee Report (RR1)

Review of *Climate and the September 2014 flood event over Mountainous Jammu and Kashmir, India: Physical explanations based on observations and modelling*

I recommend the authors spell check the document – for example there are several instances of missing spaces between words. There are a number of additional instances of poor grammar, as such I further recommend that the authors have the manuscript read by a native speaker.

L26-7: Details of statistical tests should not be in the abstract; reserve these for your methods section

L30-33: It is not conventional to state confidence ranges in this way. In the abstract it will suffice to say that the changes are significant.

L49-50: This sentence is poor English, please rewrite.

L200: What is the significance of the bar over *x* and *y*?

L208: The authors discussed in the introduction the importance of the orography and orographically-driven precipitation in the region; can they reconcile this with the fairly coarse 9-km resolution used here?

L220-226: A brief justification of these choices would be welcome (though I won't require it).

L432: "60% of surface temperature"? Do you mean 60% of its variance?

L453: You should introduce ERA-Interim and its citation in the methodology, not deep in the results

L468: "msl higher by 1 km" – what do you mean here? 1 km higher in altitude?

L529: "seal"

L540: "unusual weather changes" is a bit vague. Can you be more specific?

---

## Editor Decision (ED1)

**REVIEW  of**

"Climate and the September 2014 flood event over Mountainous Jammu and Kashmir, India: Physical explanations based on observations and modelling" By Sumira et al.

**MAJOR**

1. The presentation quality has to be improved, as I (and both referees) mentioned previously.

2. The title is still not suitable for the content of the article. I would suggest something like: "Analyses of temperature and precipitation in the Indian Jammu-Kashmir for the 1980—2016 period: Implications for remote influence and extreme events"

3. "respectively" is used at many places without "respective" cases.
4. Fix the confidence interval you wanted to discuss and then just say "significant" or "insignificant". Otherwise, it would be confusing if you use S=0.01 and S=0.05 (line 30, 31-33, and throughout the article)
5. Line 184 and many other places you have used "climatic variations". What are these climatic variations? How do you find the changes or variability in climate without analyzing the temperature, precipitation or related parameters?
6. Section 4.3: In this section you have only used NAO. So how would you know that this is the main reason for the variability in temperature or precipitation?
7. You have also used "climatic" at many places for "climate"
8. You have used three pages to describe the PV and temperature/rainfall correlation analyses. The sections need to be shortened, as there are no new results. Perhaps, you could describe the connection for a station in detail and then discuss how other station measurements differ from the former.

**MINOR**

Line 19: what are these climate settings? Be specific.

Line 21: short-tern (similarly long-term)

Line 36: "show"

Line 38-39: What are the reasons for the difference?

Line 22-24: Just write that you have used measurements from six stations in the valley and long-term simulations from the WRF model.  That complex sentence is very hard to comprehend.

Line 51: climate change

Line 52: in the coastal

Line 55: "and (4)"

Line 54: "significant" delete "the"

Line 56, 59: remove etc.

Line 67, 68: remove PT when you mention potential temperature surfaces, as you have already mentioned "K" there. Use "e.g. 350 K"

Line 69: PV is conserved everywhere at all times?

Line 77—78: western disturbances and monsoon. Else, write all those "etc" things there. There are several "etc" in this manuscript, which is not a good "word" choice. If you know the related processes write those or give some examples.

Line 78: the vertical distribution

Line 79-80: please rephrase the sentence, very difficult to understand this now

Line 84: write like, "Kashmir floods in 2014, Leh floods in 2010"

Line 87: and distinguish

Line 94: PV is considered as Rossby wave activity?

Line 100: "Jain (2009)"

Line 106: that minimum

Line 112: what are these hydrometeorological parameters? Be specific.

Line 114: data pose

Line 115—116: climatic variations in weather parameters? Please write the change in that particular parameter, instead of that phrase.

Line 117:  are necessary for understanding

Line 119: climate of the Himalaya

Line 127:  ", respectively" and similarly at other places

Line 141: hard to comprehend this sentence "to be clarified with NAO only"

Line 144:  not the connection between NAO INDEX and precipitation, but NAO and precipitation. Index is just a number or set of numbers. Rewrite similar statements elsewhere in the article.

Line 161: write "3.1 Measurements and Model simulations"

Line 163: Please write "The data are analysed carefully…."

Line 170: Space before ". Trends…"

Line 184: This is not correct. You are just correlating the NAO Index with temperature and precipitation. There is no "climatic variations of mean temperature" unless you define it.

Line 221-222: Is this the best parameterization scheme for the Indian region?

Line 225: Delete "Priyanka", Just Ghosh et al.

Line 231: What are "climate data"? Specify the data here.

Line 287, 290: "insignificant level" for confidence intervals. You need to define that with respect to your estimates (e.g. 95% CI). Just write that they are "statistically insignificant".

Line 296: Influence of North Atlantic Oscillation

Line 306-307: This is not a result, but a guess. If you find something significant from your study, please indicate that clearly. "The analyses show significant correlation between NAO Index and precipitation; indicating a possible connection between NAO and rainfall in the Kashmir, where the positive correlation suggests ….." something like this.

Line 309: "at all stations..."

Line 310: downscaled

Line 311: "0.47" is good correlation?

Line 317-318: "A detailed study on this topic will be presented in a separate paper." Write something similar.

Line 334—335: write something this sort: "This is consistent with the model results of Wiltshore (2013) ……"

Line 340-341: give a reference for this statement

Line 357: Please note the study period also, as the trends depend on the length (period of analyses) of the data sets, and then compare trend during similar periods.

Line 362: Give reference for this statement.

Line 396: "Furthermore"

Line 407: "propagate"

Line 408: delete "clear from studies". This may not be very clear for everyone. Write "This study shows that" or "suggested" or use similar wordings

Line 412: (e.g. Screen and Simmonds, 2014)

Line 412: In northern India

Line 424: They also note that

Line 450: delete "PT" and at other places too

Line 453: "Therefore, PV in the upper troposphere varies"

Line 455-456: What are these sometimes and other times? years? Be specific here. Rephrase the sentence without "One can"

Line 465—4666: I am bit confused about the statement. How can you state that the particular correlation between PV and rainfall in 1999—2000 is just due to climate change? There was no effect of this climate change in other years? What do you mean by climate change here? I thought you are analyzing the changes in climate of the region in this study.

Line 477-478: The measurements show that the stations located near the Greater Himalaya…

Line 485: Equatorward

Line 486: "at both" and "except for January—March "

Line 502-503: The observations show that the high altitude mountains …

Line 510: the connection between PV and precipitation …

Line 512: PV and precipitation

Line 518: conditions such as local convection.

Line 530: subtropical

Line 532/535: there is no mechanism to "attract" the moisture there. Use "transport" instead

Line 539: what is "drifting climatically"?

Line 547: What is "climatic increase in tropospheric warming"?

Line 548: space before the bracket

Line 549: Space after the full-stop

Line 550: combined effect of

Line 554: again "climatically shifted"?

Look for corrections at other places ….

---

## Author Response (AR2)

Responses to the reviewer comments of our revised manuscript (acp – 2018-201) titled "*Climate and the September 2014 flood event over Mountainous Jammu and Kashmir, India: Physical explanations based on observations and modelling*" By Sumira et al. submitted for possible publication in the journal *Atmospheric Chemistry and Physics,* Copernicus publications.

**General response:**

We express our sincere thanks to the reviewer for his invaluable constructive comments of our manuscript. His encouraging comments helps us a lot to improve the presentation quality of the manuscript. Below, we have provided our one to one responses to the reviewing comments. We hope that the present revised manuscript, written taking into account of all the reviewer comments, will convince the reviewer to make positive recommendations of our manuscript for publication in ACP. Manuscript with changes tracked also enclosed in the end for quick reference.

**One to one responses:**

**The responses are in bold faceted.**

**MAJOR**

1. The presentation quality has to be improved, as I (and both referees) mentioned previously.

   **The quality of the manuscript is now significantly improved.**

2. The title is still not suitable for the content of the article. I would suggest something like: "Analyses of temperature and precipitation in the Indian Jammu-Kashmir for the 1980—2016 period: Implications for remote influence and extreme events"

**The title of the manuscript is now changed as suggested by the reviewer.**

3. "respectively" is used at many places without "respective" cases.

   **Carefully avoided now all through the manuscript.**

4. Fix the confidence interval you wanted to discuss and then just say "significant" or "insignificant". Otherwise, it would be confusing if you use S=0.01 and S=0.05 (line 30, 31-33, and throughout the article)

   **The present revised manuscript has taken well this into account.**

5. Line 184 and many other places you have used "climatic variations". What are these climatic variations? How do you find the changes or variability in climate without analyzing the temperature, precipitation or related parameters?

   **Unnecessary usage of "climatic" is now removed**

6. Section 4.3: In this section you have only used NAO. So how would you know that this is the main reason for the variability in temperature or precipitation?

**Enough care has been taken now to rephrase some sentences so that ambiguous statements are removed.**

7. You have also used "climatic" at many places for "climate"

**Unnecessary usage of "climatic" is now removed**

8. You have used three pages to describe the PV and temperature/rainfall correlation analyses. The sections need to be shortened, as there are no new results. Perhaps, you could describe the connection for a station in detail and then discuss how other station measurements differ from the former

**Efforts are applied to shorten this section in the revised manuscript.**

**MINOR**

Line 19: what are these climate settings? Be specific. **Modified**
Line 21: short-tern (similarly long-term) **Corrected**
Line 36: "show" **corrected**
Line 38-39: What are the reasons for the difference? **Explained**
Line 22-24: Just write that you have used measurements from six stations in the valley and long-term simulations from the WRF model. That complex sentence is very hard to comprehend. **Modified accordingly**
Line 51: climate change **Corrected**
Line 52: in the coastal **Corrected**
Line 55: "and (4)" **Corrected**
Line 54: "significant" delete "the" **Corrected**
Line 56, 59: remove etc. **Corrected**
Line 67, 68: remove PT when you mention potential temperature surfaces, as you have already mentioned "K" there. Use "e.g. 350 K" **Corrected**
Line 69: PV is conserved everywhere at all times? **Modified with additional phrases**
Line 77—78: western disturbances and monsoon. Else, write all those "etc" things there. There are several "etc" in this manuscript, which is not a good "word" choice. If you know the related processes write those or give some examples. **Corrected**
Line 78: the vertical distribution **Corrected**
Line 79-80: please rephrase the sentence, very difficult to understand this now **Modified**
Line 84: write like, "Kashmir floods in 2014, Leh floods in 2010" **Corrected accordingly**
Line 87: and distinguish **Corrected**
Line 94: PV is considered as Rossby wave activity? **It is a measure of Rossby wave activity, which is already mentioned in the text.**

Line 100: "Jain (2009)" **Corrected**
Line 106: that minimum **Corrected**
Line 112: what are these hydrometeorological parameters? Be specific. **Corrected**
Line 114: data pose **Corrected**
Line 115—116: climatic variations in weather parameters? Please write the change in that particular parameter, instead of that phrase. **Corrected**
Line 117: are necessary for understanding **Modified**
Line 119: climate of the Himalaya **Corrected**
Line 127: ", respectively" and similarly at other places **Corrected**
Line 141: hard to comprehend this sentence "to be clarified with NAO only"

   **The sentences here are modified so that it can be now comprehended easily.**

Line 144: not the connection between NAO INDEX and precipitation, but NAO and precipitation. Index is just a number or set of numbers. Rewrite similar statements elsewhere in the article. **Rewrited as suggested**

Line 161: write "3.1 Measurements and Model simulations"
**It is now written as per the reviewer suggestion**
Line 163: Please write "The data are analysed carefully…." **Corrected**
Line 170: Space before ". Trends…" **Corrected**
Line 184: This is not correct. You are just correlating the NAO Index with temperature and precipitation. There is no "climatic variations of mean temperature" unless you define it. **Modified accordingly**
Line 221-222: Is this the best parameterization scheme for the Indian region?

**We don't claim here that this is the best parameterization and already cited earlier reference which adopted similar schemes as in the present manuscript.**

Line 225: Delete "Priyanka", Just Ghosh et al. **Corrected**
Line 231: What are "climate data"? Specify the data here. **Corrected**
Line 287, 290: "insignificant level" for confidence intervals. You need to define that with respect to your estimates (e.g. 95% CI). Just write that they are "statistically insignificant". **Corrected**
Line 296: Influence of North Atlantic Oscillation **Corrected**
Line 306-307: This is not a result, but a guess. If you find something significant from your study, please indicate that clearly. "The analyses show significant correlation between NAO Index and precipitation; indicating a possible connection between NAO and rainfall in the Kashmir, where the positive correlation suggests ….." something like this. **Rewritten accordingly**
Line 309: "at all stations..." **Corrected**
Line 310: downscaled **Corrected**
Line 311: "0.47" is good correlation? **Modified this sentence**
Line 317-318: "A detailed study on this topic will be presented in a separate paper." Write something similar. **The sentence is rewritten as per the advice of the reviewer**
Line 334—335: write something this sort: "This is consistent with the model results of

Wiltshore (2013) ……" **The sentence is rewritten accordingly as suggested by the reviewer**

Line 340-341: give a reference for this statement

**Now Reference Barnes et al., 2016 is added**

Line 357: Please note the study period also, as the trends depend on the length (period of analyses) of the data sets, and then compare trend during similar periods.

**Now comparision has been done with information of study period.**

Line 362: Give reference for this statement. **It is now provided**

Line 396: "Furthermore" **Corrected**

Line 407: "propagate" **Corrected**

Line 408: delete "clear from studies". This may not be very clear for everyone. Write "This study shows that" or "suggested" or use similar wordings **Corrected**

Line 412: (e.g. Screen and Simmonds, 2014) **e.g. is now added**

Line 412: In northern India **Corrected**

Line 424: They also note that **Corrected**

Line 450: delete "PT" and at other places too **Corrected at all places**

Line 453: "Therefore, PV in the upper troposphere varies" **Corrected accordingly**

Line 455-456: What are these sometimes and other times? years? Be specific here. Rephrase the sentence without "One can" **It is now modified**

Line 465—4666: I am bit confused about the statement. How can you state that the particular correlation between PV and rainfall in 1999—2000 is just due to climate change? There was no effect of this climate change in other years? What do you mean by climate change here? I thought you are analyzing the changes in climate of the region in this study. **Rephrased this sentence**

Line 477-478: The measurements show that the stations located near the Greater Himalaya… **Corrected**

Line 485: Equatorward **Corrected**

Line 486: "at both" and "except for January—March "**Corrected**

Line 502-503: The observations show that the high altitude mountains … **Corrected**

Line 510: the connection between PV and precipitation … **Corrected**

Line 512: PV and precipitation **Corrected**

Line 518: conditions such as local convection. **Corrected**

Line 530: subtropical **Corrected**

Line 532/535: there is no mechanism to "attract" the moisture there. Use "transport" Instead **Transported replaced attracted**

Line 539: what is "drifting climatically"? **The centere of the Subtropical jet which is now mentioned**

Line 547: What is "climatic increase in tropospheric warming"? **It is long-term variation**

Line 548: space before the bracket **Corrected**

Line 549: Space after the full-stop **Corrected**

Line 550: combined effect of **Corrected**

Line 554: again "climatically shifted"? **Corrected**

Look for corrections at other places …. **Corrections are done all through the manuscript.**

[revised manuscript text omitted]
 is considered as winter season as per the IMD definition. This is because, for the NAO index, normally December-March is considered as northern winter and we adopted the same definition here (Archer and Fowler, 2004; Iqbal and Kashif, 2013). The result of linkage between winter NAO index and Kashmir precipitation does not affect other results of the present work. The annual temperature in the valley varies from about -10℃ to 35℃. The rainfall pattern in the valley is dominated by winter time precipitation associated with western disturbances (Dar et al., 2014) while the snow precipitation is received mainly in winter and early spring season (Kaul and Qadri, 1979).

**2. Data and Methodology**

India Meteorological Department (IMD) provided 37 years (1980-2016) of data of daily precipitation, maximum temperature and minimum temperatures for all these six stations. Monthly averaged data were further analysed to find long-term variations of the weather parameters. Statistical tests including Mann-Kendall, Spearman Rho, Cumulative deviation, Student's t-test were performed to determine long term-trends and turning point of weather parameters with statistical significances. Similar analyses and tests were performed also for the Weather and Research Forecasting (WRF) model simulated and ERA-Interim reanalyses data (0.75° by 0.75° spatial resolution in the horizontal plane, monthly averaged time resolution) of same weather parameters and for the NAO index. Brief information about these data sets is provided below.

**3.1 Measurements and model simulationsObservational and model datasets used in this study**

The obtained observational data arewere analysed carefully analysed for homogeneity and missing values. Analyses of ratios of temperature from the neighbouring stations with the Srinagar station were conducted using relative homogeneity test (WMO, 1970). It is found that there is no significant inhomogeneity and data gap for any station. Few missing data points were linearly interpolated and enough care was taken not to make any meaningful interpretation during such short periods of data gaps in the observations. Annual and seasonal means of temperature and precipitation were calculated for all the stations and years. To compute seasonal means, the data were divided into the following seasons: winter (December to February), spring (March to May), summer (June to August) and autumn (September to November). Trends in the annual and seasonal means of temperature and precipitation were determined using Mann–Kendall (non-parametric test) and linear regression tests (parametric test) at the confidence levels of S = 99% or (0.01), S = 95% or 0.05 and S = 90% or 0.1. These tests have been extensively used in hydro meteorological data analyses as they are less sensitive to heterogeneity of data distribution and least affected by extreme values or outliers in data series. Various methods have been applied to determine change points of a time series (Radziejewski et al., 2000; Chen and Gupta, 2012). In this study, change point in time series of temperature and precipitation was identified using cumulative deviation test and Student's t test (Pettitt, 1979). This method detects the time of significant change in the mean of a time series when the exact time of the change is unknown (Gao et al., 2011).

winter NAO index during 1980–2010 were obtained for further analyses from Climatic

Research Unit  through the web link https//www.cru.uea.ac.uk/data. The winter (December - March) NAO index is based on  difference of normalized sea level pressure (SLP) between Lisbon,

Portugal and Iceland, which is available from 1964 onwards. Positive NAO index is associated with stronger-than- average westerlies over the middle latitudes (Hurrell, 1997). Correlation between  mean (December-March) temperature, precipitation and NAO index was determined using Pearson correlation coefficient method. To test whether the observed trends in winter temperature and precipitation are enforced by NAO, linear regression analysis (forecast) was performed (Fig. 4e and f). The following algorithm calculates or predicts a future value by using existing values. The predicted value is a y-value for a given w-value. The known values are existing w-values and y- values, and the new value is predicted by using linear regression.

The syntax is as follows
FORECAST(x, known_y's, known_w's)
W   is the data point for which we want to predict a value.

Known_y's   is the dependent array or range of data (rainfall or temperature).

Known_w's   is the independent array or range of data (time).

The equation for FORECAST is a + bw, where:

$$a = \hat{y} - b\hat{w} \quad \text{and} \quad b = \sum(w\text{-}\hat{w})(y\text{-}\hat{y})/\sum(w\text{-}\hat{w})^2$$

$$a = \bar{y} - b\bar{x}$$

[revised manuscript text omitted]

Font: (Default) Times New Roman, 10 pt, Complex Script Font: Times New Roman, 10 pt

| Page 23: [2] Formatted | RAMKUMAR | 11/16/2018 3:00:00 PM |
|---|---|---|

Font: (Default) Times New Roman, 10 pt, Complex Script Font: Times New Roman, 10 pt

---

## Author Response (AR3)

Responses to the Co-Editor Prof. Jayanarayanan Kuttippurath for his important suggestions of our final stage manuscript (acp – 2018-201) titled "*Climate and the September 2014 flood event over Mountainous Jammu and Kashmir, India: Physical explanations based on observations and modelling*" by Sumira et al. submitted for publication in the journal *Atmospheric Chemistry and Physics,* Copernicus publications.

**General response:**

We are indebted to the Co-Editor Prof. Jayanarayanan Kuttippurath for his important suggestions about improving the quality of the figures, adding information about trends in the rainfall and temperature in the abstract and correcting typos in the final stage of this manuscript. We hope that the Co-Editor will be convinced of this final stage manuscript. Manuscript with changes tracked version also enclosed for quick reference. If any further improvement is needed for figures, we will surely make our sincere attempts so that ACP journal quality figures will be produced.

**One to one responses:**

**The responses are in bold faceted.**

1. The figures are of very poor quality. Please improve the quality of the figures for the final upload. Figures 1-6, the font size of the axes are too small to read. Please consider to re-draw them. This is a very important study for that region. Therefore, please take care all aspects of the manuscript. Thank you.

**The quality of all the figures is improved now and if any further improvement is needed, we will surely do it during production stages for ACP publication quality figures.**

2. Abstract: You have mentioned the increase and decrease in precipitation, please provide the trend values with uncertainty there. Also, add uncertainty values (error range or stddev) with temperature trends.

**The following sentences are now added**

**In the present study, the observed long-term trends in temperature (°C/year) and precipitation (mm/year) along with their respective standard errors during 1980-2016 are as follows: (1) 0.05 (0.01) and -16.7 (6.3) for Gulmarg, (2) 0.04 (0.01) and -6.6 (2.9) for Srinagar, (3) 0.04 (0.01) and -0.69 (4.79) for Kokernag, (4) 0.04 (0.01) and -0.13 (3.95) for Pahalgam, (5) 0.034 (0.01) and -5.5 (3.6) for Kupwara and (6) 0.01 (0.01) and -7.96 (4.5) for Quazigund.**

3. There are still some technical errors, but this would be taken care of during the proof reading. However, if you can correct some, correct those to speed-up the publication process.
e.g. Line 288, line 331, line 454,

**All typos are now corrected.**

**Analyses of temperature and precipitation in the Indian Jammu-Kashmir for the 1980—2016 period: Implications for remote influence and extreme events**

Sumira Nazir Zaz[1], Romshoo Shakil Ahmad[1], Ramkumar Thokuluwa Krishnamoorthy[2*], and Yesubabu Viswanadhapalli[2]

1. Department of Earth Sciences, University of Kashmir, Hazratbal, Srinagar, Jammu and Kashmir-190006, India

2. National Atmospheric Research Laboratory, Dept. of Space, Govt. of India, Gadanki, Andhra Pradesh 517112, India

**Email:** zaz.sumira@gmail.com, shakilrom@kashmiruniversity.ac.in, tkram@narl.gov.in, yesubabu@narl.gov.in;

*Corresponding author** (tkram@narl.gov.in)

**Abstract**

Local weather and climate of the Himalayas are sensitive and interlinked with global scale changes in climate as the hydrology of this region is mainly governed by snow and glaciers. There are clear and strong indicators of climate change reported for the Himalayas, particularly the Jammu and Kashmir region situated in the western Himalayas. In this study, using observational data, detailed characteristics of long- and short-term as well as localized variations of temperature and precipitation are analysed for these six meteorological stations, namely, Gulmarg, Pahalgam, Kokarnag, Qazigund, Kupwara and Srinagar of Jammu and Kashmir, India during 1980-2016. In addition to analysis of stations observations, we also utilized the dynamical downscaled simulations of WRF model and ERA-Interim (ERA-I) data for the study period. The annual and seasonal temperature and precipitation changes were analysed by carrying out Student's t-test, Mann-Kendall, Linear regression and Cumulative deviation statistical tests. The results show an increase of 0.8°C in average annual temperature over thirty seven years (from 1980 to 2016) with higher increase in maximum temperature (0.97°C) compared to minimum temperature (0.76°C). Analyses of annual mean temperature at all the stations reveal that the high-altitude stations of Pahalgam (1.13°C) and Gulmarg (1.04°C) exhibit a steep increase and statistical significant trends. The overall pPrecipitation patterns in the valley such as at Gulmarg and Pahalgam shows significanta slight and definite decrease in the annual rainfall. at Gulmarg and Pahalgam stations. Seasonal analyses show significant increasing trends in the winter and spring temperatures at all stations with prominent decrease in spring precipitation. In the present study, the observed long-term trends in temperature (°C/year) and precipitation (mm/year) along with their respective standard errors during 1980-2016 are as follows: (1) 0.05 (0.01) and -16.7 (6.3) for Gulmarg, (2) 0.04 (0.01) and -6.6 (2.9) for Srinagar, (3) 0.04 (0.01) and -0.69 (4.79) for Kokernag, (4) 0.04 (0.01) and -0.13 (3.95) for Pahalgam, (5) 0.034 (0.01) and -5.5

[revised manuscript text omitted]